# Excessive reactive oxygen species induce transcription-dependent replication stress

Martin Andrs[1,2], Henriette Stoy [2], Barbora Boleslavska[1,3], Nagaraja Chappidi[2,7], Radhakrishnan Kanagaraj [4,5,6], Zuzana Nascakova[1,8], Shruti Menon[2,9], Satyajeet Rao[2], Anna Oravetzova[1,3], Jana Dobrovolna [1], Kalpana Surendranath[4,6], Massimo Lopes [2] & Pavel Janscak [1,2] ✉

Elevated levels of reactive oxygen species (ROS) reduce replication fork velocity by causing dissociation of the TIMELESS-TIPIN complex from the replisome. Here, we show that ROS generated by exposure of human cells to the ribonucleotide reductase inhibitor hydroxyurea (HU) promote replication fork reversal in a manner dependent on active transcription and formation of co-transcriptional RNA:DNA hybrids (R-loops). The frequency of R-loop-dependent fork stalling events is also increased after TIMELESS depletion or a partial inhibition of replicative DNA polymerases by aphidicolin, suggesting that this phenomenon is due to a global replication slowdown. In contrast, replication arrest caused by HU-induced depletion of deoxynucleotides does not induce fork reversal but, if allowed to persist, leads to extensive R-loop-independent DNA breakage during S-phase. Our work reveals a link between oxidative stress and transcription-replication interference that causes genomic alterations recurrently found in human cancer.

DNA replication stress is a pathologic condition characterized by slowing or stalling of DNA replication fork progression, caused either by obstacles such as DNA lesions, secondary DNA structures or active transcription complexes, or, alternatively, by a reduction in the cellular pools of deoxynucleotide triphosphates (dNTPs)[1]. In most studies to date, DNA replication stress has been induced by the ribonucleotide reductase (RNR) inhibitor hydroxyurea (HU), which causes dNTP depletion[2], replication fork arrest and, ultimately, DNA breakage[3–5]. However, exposure of cells to HU also leads to generation of reactive oxygen species (ROS) and activation of the oxidative stress response even at 50 μM concentration that is too low to bring about dNTP depletion[6–10]. Interestingly, elevated ROS levels have been shown to disrupt the oligomeric state of the replisome-associated ROS sensor peroxiredoxin 2 (PRDX2), thereby enforcing the displacement of the

TIMELESS-TIPIN complex from the replisome, which globally reduces the rate of replication fork progression[10]. HU, as well as hydrogen peroxide (H$_2$O$_2$), are also known to actively interfere with replication fork progression by inducing replication fork reversal, a DNA transaction mediated by RAD51 recombinase and the DNA translocases ZRANB3, HLTF and SMARCAL1[5,11,12]. Notably, exposure of cells to 50 μM HU gives rise to a similar number of reverse forks as do >500 μM HU concentrations that cause dNTP depletion[5,10,11]. However, how replication slowdown can induce fork reversal remains unclear.

Our recent study showed that one cause of replication fork reversal is replication blockage by a co-transcriptional R-loop[13], a secondary DNA structure generated by invasion of the nascent transcript into the underwound DNA duplex behind the transcription complex, which leads to the formation of an RNA:DNA hybrid on the template

[1]Institute of Molecular Genetics of the Czech Academy of Sciences, Prague, Czech Republic. [2]Institute of Molecular Cancer Research, University of Zurich, Zurich, Switzerland. [3]Faculty of Science, Charles University in Prague, Prague, Czech Republic. [4]Genome Engineering Laboratory, School of Life Sciences, University of Westminster, London, UK. [5]School of Life Sciences, University of Bedfordshire, Luton, UK. [6]Centre for Drug Discovery and Development, Sathyabama Institute of Science and Technology, Chennai, India. [7]Present address: Center for Molecular and Cellular Bioengineering, Biotechnology Center, Technische Universität Dresden, Dresden, Germany. [8]Present address: Institute of Molecular Cancer Research, University of Zurich, Zurich, Switzerland. [9]Present address: School of Medicine, University of California San Francisco (UCSF), San Francisco, CA, USA. ✉e-mail: pjanscak@imcr.uzh.ch

DNA strand[14]. R-loop formation is promoted by head-on transcription-replication conflicts (TRCs) and is favored in the regions containing G-quadruplex (G4)-forming sequences in the non-transcribed DNA strand[14–17]. Here, we provide evidence that ROS-induced replication fork slowing leads to replication fork reversal due to interference with transcription that is associated with R-loop formation. Moreover, we demonstrate that replication arrest induced by dNTP depletion does not induce fork reversal per se but, if allowed to persist, causes R-loop-independent DNA breakage during S-phase.

## Results

### ROS-induced replication slowdown causes replication fork stalling

In the absence of replication fork protection factors such as BRCA1 and BRCA2, reversed replication forks in HU-treated cells become vulnerable to MRE11 nuclease, leading to extensive resection of nascent DNA strands, which can be monitored at a single molecule level by labeling replication tracts with halogenated thymidine analogs and their visualization on DNA fiber spreads by indirect immunofluorescence imaging[18,19]. Consistent with a recent study[20], we found using the DNA fiber assay that nascent DNA degradation in BRCA2-depleted U2OS cells exposed to 4 mM HU for 5 h was completely attenuated by N-acetyl-L-cysteine (NAC), an efficient scavenger of ROS (Fig. 1a–d), but not by the addition of exogenous nucleosides (Fig. 1d). Furthermore, we also observed that this fork degradation phenotype of BRCA2-deficient cells was suppressed by inhibition of PRDX2 with Conoidin A or by PRDX2 knockdown (Fig. 1d; Supplementary Fig. 1a–c), which can rescue ROS-induced replication slowdown likely without affecting ROS levels in the cell[10]. This suggests that replication fork reversal, which triggers nascent DNA strand degradation in BRCA2-deficient cells[18], results from ROS-induced replication slowdown, rather than from replication arrest due to dNTP deficiency. In support of this notion, we found that extensive fork degradation took place also in the presence of a low concentration of HU (50 μM), or after exposure of cells to $H_2O_2$ (Supplementary Fig. 1e).

Interestingly, we also observed that nascent DNA tracts of sister replication forks in HU-treated BRCA2-deficient cells were degraded in an asymmetric manner (Fig. 1b, e). Similarly, in addition to a reduction in replication fork velocity, a marked asymmetry of sister replication tracts was observed in BRCA2-proficient cells after a 30-min treatment with HU (both 50 μM and 500 μM), $H_2O_2$ or buthionine sulfoximine (BSO), an endogenous ROS producer (Fig. 1f–i; Supplementary Fig. 1f–h), suggesting replication fork stalling. Importantly, these sister fork asymmetry phenotypes could be fully rescued by scavenging ROS with NAC or by inhibiting or depleting PRDX2 (Fig. 1e, i; Supplementary Fig. 1d, h, i–k), but not by adding exogenous nucleosides (Supplementary Fig. 1l–n). As expected, NAC or Conoidin A also prevented HU- or $H_2O_2$-induced replication fork slowing but, upon treatment with a high HU concentration (500 μM), this replication speed rescue was only marginal, due to the prevailing effect of dNTP shortage (Fig. 1h). The addition of exogenous nucleosides partly restored normal replication fork speed in cells exposed to 500 μM HU and had no effect on replication speed in cells treated with 50 μM HU or $H_2O_2$ (Supplementary Fig. 1n). Collectively, these results suggest that, upon oxidative stress, the global replication fork slowing due to TIMELESS-TIPIN dissociation induces "local" replication fork stalling events associated with fork reversal, which act as entry points for fork degradation in cells lacking BRCA2. In line with this hypothesis, TIMELESS depletion, mimicking its ROS-induced displacement from the replisome, also resulted in sister fork asymmetry, as previously reported[21], and this effect was not exacerbated by exposure of cells to 50 μM HU (Supplementary Fig. 1o–t). In addition, we found that abrogation of fork reversal by ZRANB3 depletion completely prevented sister fork asymmetry, but only partially restored the normal rate of fork progression in U2OS cells treated with 50 μM HU (Fig. 1j–m), further

supporting the notion that replication slowdown due to TIMELESS-TIPIN dissociation is the cause of ROS-induced fork stalling.

### ROS-induced fork stalling is caused by co-transcriptional R-loops

HU was shown to stimulate transcription-dependent breakage of specific genomic loci, termed early replicating fragile sites (ERFSs)[22]. We, therefore, sought to examine whether replication fork stalling events observed in HU- and $H_2O_2$-treated cells resulted from TRCs associated with R-loop formation. To this end, we made use of a stable U2OS T-REx cell line conditionally overexpressing RNase H1, a nuclease that eliminates R-loops[13]. We found that RNase H1 overexpression could almost completely rescue sister fork asymmetry, and partially restore normal rates of replication fork progression in HU-, $H_2O_2$- or BSO-treated cells (Fig. 2a–d; Supplementary Fig. 2a–c). The same effects were observed if U2OS or RPE-1 cells were pretreated with the transcription elongation inhibitors 5,6-dichloro-1-β-D-ribofuranosylbenzimidazole (DRB) or cordycepin (CORD), respectively, or with the transcription initiation inhibitor triptolide (TRP) prior to the addition of ROS-generating drugs (Supplementary Fig. 2d–l). In addition, we found that RNase H1 overexpression or transcription inhibition rescued sister fork asymmetry and partly restored normal replication velocity in U2OS cells depleted of TIMELESS (Fig. 2e–g; Supplementary Fig. 2m–o), and prevented HU- or $H_2O_2$-induced nascent DNA degradation in BRCA2-depleted cells (Fig. 2h–j; Supplementary Fig. 2p–s). To obtain direct evidence that HU-induced fork reversal depends upon ROS and co-transcriptional R-loops, we analyzed replication intermediates by electron microscopy following in vivo psoralen crosslinking[23]. U2OS T-REx cells conditionally expressing RNase H1 were treated with 500 μM HU for 1 h, which causes both elevation of ROS and dNTP depletion[10]. Under these conditions, reversed forks represented around 20% of replication intermediates (Fig. 2k), consistent with the fork reversal frequency observed in this cell line upon treatment with R-loop-inducing drugs such as camptothecin (CPT) or pyridostatin[13]. In contrast, the frequency of fork reversal was significantly reduced if HU treatment was carried out in the presence of NAC or DRB, or after induction of RNase H1 overexpression by doxycycline (Fig. 2k). Collectively, these results suggest that ROS-induced replication fork stalling is caused by co-transcriptional R-loops. Moreover, our data imply that HU-induced fork reversal results from R-loop-mediated TRCs caused by ROS, rather than from replication arrest due to dNTP depletion. Of note, inhibition of origin firing by the CDC7 inhibitor XL413 did not prevent replication fork stalling in cells treated with 50 μM HU (Supplementary Fig. 2t–v), which excludes the possibility that it was caused by dormant origin firing.

### Elevated ROS levels induce R-loop formation in S-phase cells in a manner dependent on PRDX2

To directly demonstrate that ROS induce persistent TRCs causing fork stalling, we measured the levels of co-localization between the DNA polymerase processivity factor PCNA and the elongating form of RNA polymerase II (RNAPII) in mock- and HU-/$H_2O_2$-treated U2OS cells, using proximity ligation assay (PLA) followed by high content microscopy and quantitative image-based cytometry (QIBC)[4]. Before the addition of ROS-generating drugs, cells we subjected to a brief pretreatment with 5-Ethynyl-2′-deoxyuridine (EdU) that allows the visualization of newly-synthesized DNA. We observed that a 1-h exposure of cells to 50 μM HU or $H_2O_2$ dramatically increased the number of PLA foci between PCNA and RNAPII in EdU-positive nuclei (Fig. 3a, b; Supplementary Fig. 3a, b). Such increase in co-localization between PCNA and RNAPII was not seen if HU was combined with NAC (Fig. 3c), suggesting that HU induces TRCs in a manner dependent on ROS.

To prove that ROS generated by HU treatment induce the formation of R-loops, we made use of a previously established U2OS

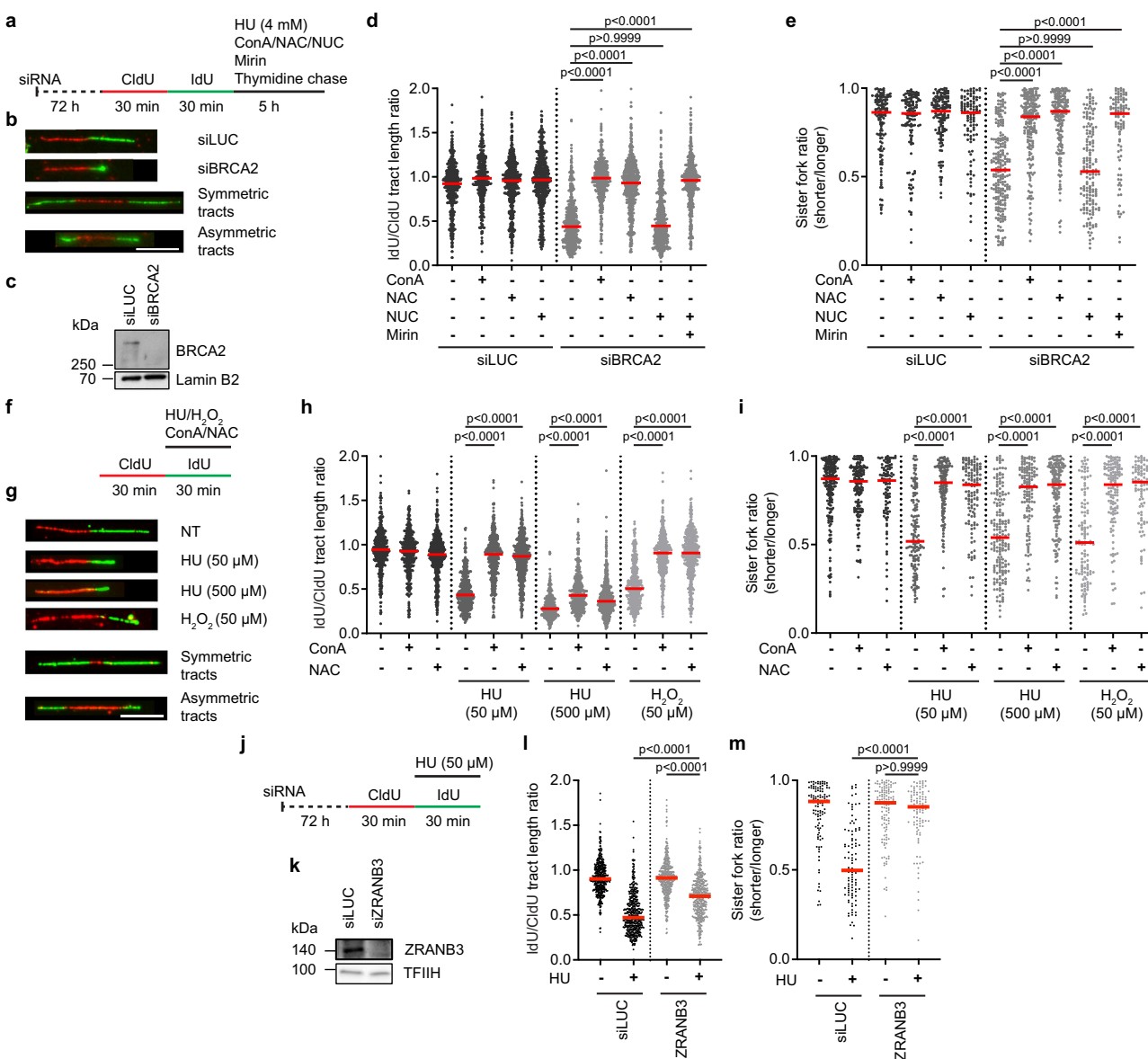

**Fig. 1 | ROS-induced replication slowdown causes replication fork stalling.**
**a**–**e** Elimination of ROS or PRDX2 inhibition prevents nascent DNA degradation in BRCA2-depleted U2OS cells upon exposure to hydroxyurea (HU). **a** Experimental workflow of DNA fiber assays. ConA, Conoidin A (PRDX2 inhibitor; 0.5 μM); NAC, N-acetyl cysteine, (ROS scavenger; 5 mM); NUC, exogenous nucleosides (adenosine, guanosine, thymidine and cytosine; 20 μM each), Mirin, (MRE11 inhibitor; 50 μM). A thymidine (400 μM) chase was included for all conditions to stop IdU incorporation. **b** Representative images of DNA replication tracts. Scale bar, 10 μm. **c** Western blot analysis of the extracts of U2OS cells transfected with control siRNA (siLuc) or siRNA to BRCA2 (siBRCA2). **d** Scatter plot of the values of IdU/CldU tract length ratio obtained from three independent experiments for indicated conditions (n ≥ 371). **e** Scatter plot of the values of sister IdU tract length ratio (sister fork ratio; shorter tract/longer tract) obtained from three independent experiments for indicated conditions (n ≥ 100). **f**–**i** ROS induce replication fork stalling in a PRDX2

dependent manner in U2OS cells. **f** Experimental workflow. ConA and NAC were present at concentrations as in (**a**). **g** Representative DNA fiber images. NT, non-treated. Scale bar, 10 μm. **h** Scatter plot of the values of IdU/CldU tract length ratio obtained from three independent experiments for indicated conditions (n ≥ 371). **i** Scatter plot of the values of sister fork ratio obtained from three independent experiments for indicated conditions (n ≥ 90). **j**–**m** ZRANB3 depletion rescues HU-induced fork stalling in U2OS cells. **j** Experimental workflow. **k** Western blot analysis of the extracts of U2OS cells transfected with indicated siRNAs. **l** Scatter plot of the values of IdU/CldU tract length ratio obtained from three independent experiments for indicated conditions (n ≥ 402). **m** Scatter plot of the values of sister fork ratio obtained from three independent experiments for indicated conditions (n ≥ 96). **d**, **e**, **h**, **i**, **l**, **m** Red horizontal lines indicate the median; p values were calculated by Kruskal-Wallis test followed by Dunn's multiple comparisons test. Source data are provided as a Source Data file.

T-REx cell line conditionally expressing catalytically-inactive RNase H1 fused to green fluorescence protein [RNH1(D210N)-GFP][24]. After induction of RNH1(D210N)-GFP expression with doxycycline, cells were subjected to HU treatment for 1 h followed by preextraction and immunofluorescence staining of PCNA to determine the DNA replication status of the individual cells. The treatment was carried out at both high (4 mM) and low (50 μM) concentrations of HU. QIBC revealed that both concentrations of HU induced the formation of nuclear foci of

RNH1(D210N)-GFP in PCNA-positive cells, indicating S-phase-specific formation R-loops (Fig. 3d–f; Supplementary Fig. 3c, d). A significant increase in the number of RNH1(D210N)-GFP foci was also observed in G2 cells upon HU treatment (Fig. 3g; Supplementary Fig. 3e), which may reflect persistent TRCs from late S-phase. Importantly, the accumulation of RNH1(D210N)-GFP foci in HU-treated cells was attenuated by the addition of NAC (Fig. 3e, f; Supplementary Fig. 3c, d), suggesting that HU induces R-loop formation in a manner dependent on ROS.

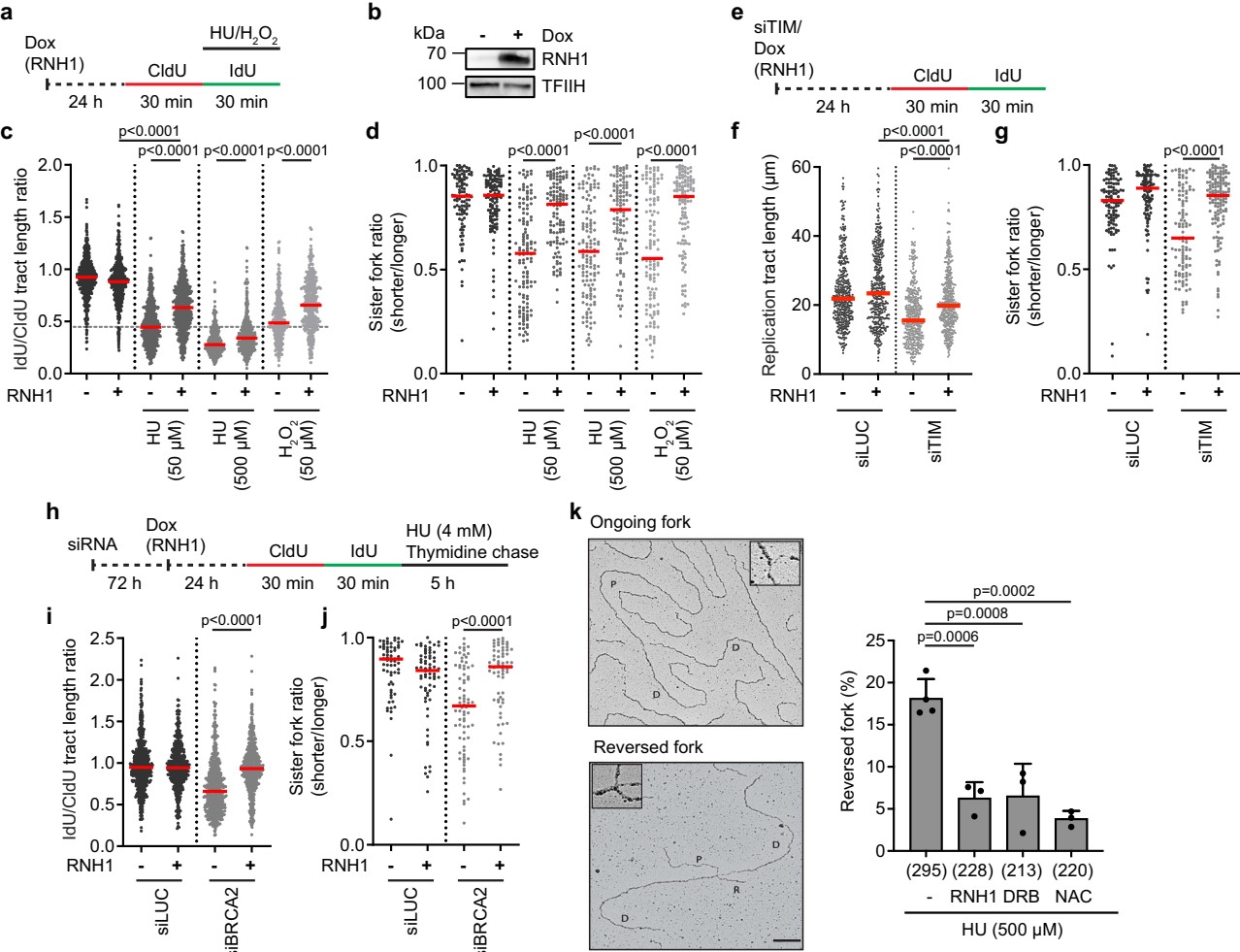

**Fig. 2 | ROS-induced fork stalling is caused by co-transcriptional R-loops.**
**a–d** Overexpression of RNase H1 prevents replication fork stalling in HU- and H$_2$O$_2$-treated cells. **a** Experimental workflow. Doxycycline (Dox; 1 ng/ml) was added to induce RNase H1 (RNH1) expression. **b** Western blot analysis of the extracts of U2OS T-REx [RNH1(WT)-GFP] cells treated with Dox for 24 h. **c** Scatter plot of the values of IdU/CldU tract length ratio obtained from three independent experiments for indicated conditions (n ≥ 319). **d** Scatter plot of the values of sister fork ratio obtained from three independent experiments for indicated conditions (n ≥ 105). **e–g** RNase H1 overexpression prevents replication fork stalling in TIMELESS-depleted cells (siTIM; 10 nM); (**e**) Experimental workflow. **f** Scatter plot of the values of replication tract length (IdU + CldU) obtained from three independent experiments for indicated conditions (n ≥ 413). **g** Scatter plot of the values of sister fork ratio obtained from four independent experiments for indicated conditions (n ≥ 106). **h–j** RNase H1 overexpression prevents HU-induced nascent DNA degradation in BRCA2-depleted U2OS cells. **h** Experimental workflow. **i** Scatter plot of the values of the IdU/CldU tract length ratio obtained from three independent experiments for indicated conditions (n ≥ 471). **j** Scatter plot of the values of sister fork ratio obtained from three independent experiments for indicated conditions (n ≥ 67). **c**, **d**, **f**, **g**, **i**, **j** Red horizontal lines indicate the median; *p* values were calculated by Kruskal-Wallis test with Dunn's multiple comparisons test. **k** HU induces fork reversal in a ROS- and R-loop-dependent manner. *Left*: Representative images of ongoing (top) and reversed (bottom) forks obtained by EM. Scale bar, 200 nm. P parental arm, D daughter arm, R regressed arm. *Right*: Quantification of the frequency of reversed forks for indicated conditions. U2OS T-REx [RNH1(WT)-GFP] cells were treated with 500 μM HU for 1 h. Dox was added 24 h before the treatment to induce RNH1 expression. DRB (100 μM) was added 2 h before HU treatment. NAC (5 mM) was added simultaneously with HU. Data are presented as mean ± SD, n = 3–4; *p* values were calculated by one-way ANOVA followed by Tukey's test. In brackets, the total number of analyzed molecules is given. Source data are provided as a Source Data file.

To substantiate these findings, we tested the effect of HU on the level of R-loops at R-loop-prone regions of the APOE, BTBD19, RPL13A, BCL2 and GADD45A genes[25–27]. DNA-RNA immunoprecipitation (DRIP) with the S9.6 antibody followed by quantitative PCR (qPCR) analysis showed that a 1-h exposure of cells to 4 mM or 50 μM HU-induced R-loop accumulation at all R-loop-prone loci tested but not at the *SNRPN* gene locus (Supplementary Fig. 3f, g), which is used as negative control[25]. Importantly, the observed R-loop accumulation at all tested loci was significantly reduced by the addition of NAC prior to HU treatment, again suggesting a dependence on ROS (Supplementary Fig. 3f, g).

A recent study has shown that locally produced ROS-induced R-loop formation in an adjacent transcribed locus, which was accompanied by the generation of transcription-independent DNA damage at

this locus revealed by accumulation of the phosphorylated form of histone H2AX (γH2AX)[28]. Based on these findings, it was proposed that ROS may promote R-loop formation by generation of DNA single-strand breaks[28]. Consistently, biochemical experiments demonstrated that a nick on the non-template DNA strand can serve as a strong R-loop initiation site[29]. However, it is not likely that such a mechanism plays a major role in HU-induced R-loop formation since the exposure of cells to 50 μM HU did not increase γH2AX levels in U2OS cells[10]. Moreover, we found that accumulation of RNH1(D210N)-GFP foci in HU-treated cells was suppressed by PRDX2 depletion (Fig. 3h; Supplementary Fig. 3h–j). In addition, PRDX2 depletion prevented formation of PLA foci between PCNA and RNAPII after exposure of U2OS cells to 50 μM HU (Fig. 3i). These findings suggest that HU-induced R-loops might form as a consequence of ROS-induced replication

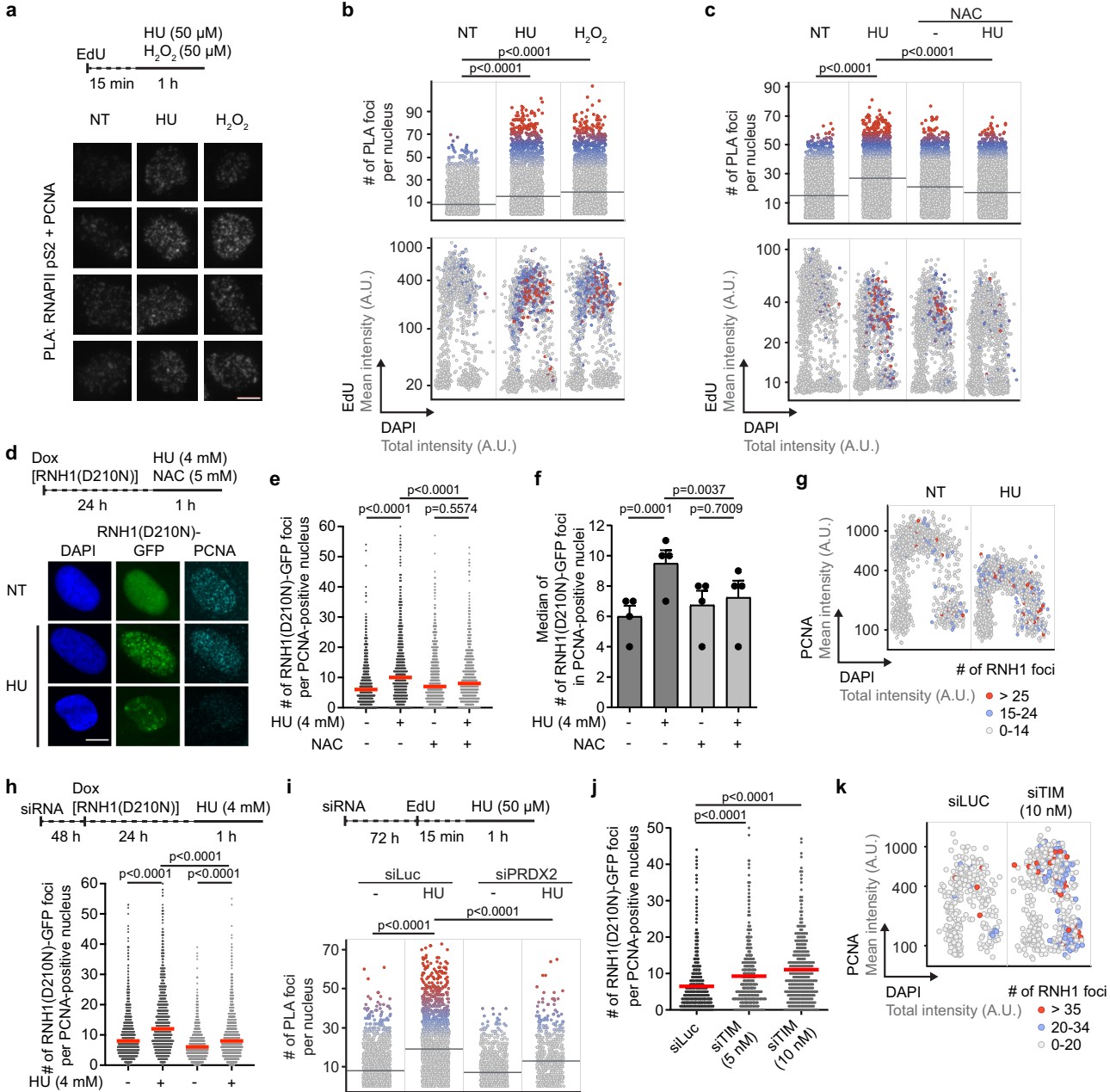

**Fig. 3 | ROS induce transcription-replication conflicts and R-loop formation in a manner dependent on PRDX2. a, b** Co-localization of PCNA and elongating form of RNA polymerase II (RNAPII pS2) in U2OS cells after 1-h exposure to HU or H₂O₂ as determined by proximity ligation assays (PLA). **a** *Top*: Experimental workflow. EdU, 5-ethynyl-2'-deoxyuridine (10 μM). *Bottom*: Representative galleries of PLA signal in cell nuclei exported from the ScanR analysis program. Scale bar, 10 μm. **b** *Top*: Scatter plot of the number of PLA foci in individual cells for indicated conditions (*n* ≥ 1508). *Bottom*: Scatter plot of total DAPI (x-axis) and mean EdU (y-axis) intensities in individual cells. Colors indicate the number of PLA foci and correspond to the upper plot. Representative plots from three independent experiments yielding similar results are shown. NT non-treated, A.U. arbitrary units. **c** HU-induced co-localization between PCNA and elongating form of RNAPII depends on ROS. The plots shown are as in (**b**) (*n* ≥ 2607). U2OS cells were treated as in (**a**). N-acetyl cysteine (NAC; 5 mM) was present during the HU treatment. **d**–**g** ROS-dependent formation RNH1(D210N)-GFP foci upon exposure of U2OS T-REx [RNH1(D210N)-GFP] cells to 4 mM HU for 1 h. **d** *Top*: Experimental workflow. *Bottom*: Representative images of DAPI, RNH1(D210N)-GFP, and PCNA channels for indicated conditions. Scale bar, 10 μm. **e** Scatter plot of the number of RNH1(D210N)-GFP foci

in PCNA-positive nuclei for indicated conditions (*n* ≥ 896). A representative plot from four independent experiments yielding similar results is shown. **f** Plot of the median values of the data sets represented in (**e**). Data are presented as mean ± SEM, *n* = 4; *p* values were calculated by one-way ANOVA followed by Tukey's test. **g** Scatter plot of DAPI total (x-axis) and PCNA mean (y-axis) intensities in individual cells. Colors indicate the number of RNH1(D210N)-GFP foci, as shown in the legend on the right. **h** HU-induced accumulation of RNH1(D210N)-GFP foci in S-phase cells depends on PRDX2. *Top*: Experimental workflow. *Bottom*: Scatter plot as in (**e**) for indicated conditions (*n* ≥ 1097). **i** HU-induced co-localization between PCNA and elongating form of RNAPII depends on PRDX2. *Top*: Experimental workflow. *Bottom*: Scatter plot as in (**b**) for indicated conditions (*n* ≥ 558). A representative plot from two independent experiments yielding similar results is shown. **j, k** TIMELESS depletion induces the formation of RNH1(D210N)-GFP foci in S-phase nuclei of U2OS T-REx [RNH1(D210N)-GFP]. **j** as in (**e**) for indicated conditions (*n* ≥ 370). **k** as in (**g**) for indicated conditions. **b, c, e, h, i, j** Horizontal lines indicate the median; *p* values were calculated by Kruskal-Wallis test followed by Dunn's multiple comparisons test. Source data are provided as a Source Data file.

slowdown and not due to the presence of ROS per se. Consistently, we also observed an accumulation of RNH1(D210N)-GFP foci in cells depleted of TIMELESS (Fig. 3j, k; Supplementary Fig. 3k, l).

## MUS81-LIG4-PRIMPOL axis mediates replication restart following ROS-induced fork stalling

We recently showed that R-loop-induced replication fork reversal is followed by the restart of semiconservative DNA replication mediated by RECQ5 DNA helicase, MUS81-EME1 endonuclease, the DNA ligase IV (LIG4)/XRCC4 complex, the transcription elongation factor ELL and the non-catalytic subunit of DNA polymerase delta, POLD3[13]. We, therefore, investigated whether these proteins mediate replication restart after ROS-induced fork stalling. By DNA fiber assay, we found that depletion of either of these proteins prevented the partial restoration of fork progression induced by depletion of ZRANB3 or HLTF translocases in U2OS cells treated with 50 μM HU (Fig. 4a, b; Supplementary Fig. 4a, b). This unrestrained DNA synthesis also required the primase-polymerase PRIMPOL, as previously reported[5], but not RECQ1 DNA helicase (Fig. 4a, b; Supplementary Fig. 4a, b), which eliminates reversed forks to promote replication restart[30]. Impaired replication fork progression in HU-treated cells could also be partially rescued by PARP inhibition with olaparib (Supplementary Fig. 4c), which promotes the immediate restart of reversed forks by RECQ1-mediated reverse branch migration[30]. As in the case of fork reversal abrogation, this olaparib-stimulated DNA synthesis in HU-treated cells was also dependent on the above proteins and, as expected, additionally required RECQ1 (Supplementary Fig. 4c). Finally, we tested whether the above-listed proteins were necessary for the restart of DNA replication following exposure of U2OS cells to 2 mM HU, which causes replication arrest. For this, HU was added to cells at the end of the first pulse labeling with CldU and incubation was continued for 2 h followed by IdU labeling in a drug-free medium (Fig. 4c). This allowed us to monitor persistent fork stalling by measuring the percentage of CldU tracts without an IdU tract (Fig. 4c). We found that cells depleted of any of the proteins including MUS81, RECQ1, RECQ5, LIG4, ELL, POLD3 and PRIMPOL displayed a marked increase in the frequency of stalled forks after HU treatment as compared to mock-depleted cells (Fig. 4d), suggesting a replication restart defect. PRIMPOL knockdown did not further increase the percentage of stalled forks in MUS81 knockout cells after release from HU arrest (Fig. 4e–g), suggesting that MUS81 and PRIMPOL act in the same replication restart pathway. Importantly, MUS81 depletion did not abolish the restart of HU-stalled forks if HU treatment was carried out in the presence of NAC (Fig. 4h). Taken together, these results provide further evidence that HU induces R-loop-mediated fork stalling in a ROS-dependent manner and establish PRIMPOL as a component of the MUS81-LIG4 axis that mediates replication restart following fork stalling by R-loops.

Interestingly, we found that MUS81 and PRIMPOL, but not RECQ1, were also necessary for the partial restoration of fork progression by RNase H1 overexpression in cells treated with 50 μM HU (Fig. 4i, j). On the other hand, replication fork slowing induced by CPT, which promotes R-loop formation by inhibiting Top1[13,31], was rescued by RNase H1 overexpression in a MUS81/PRIMPOL-independent manner (Supplementary Fig. 4d). Moreover, neither MUS81 nor PRIMPOL were required for the partial rescue of fork progression by inhibition of transcription in HU-treated cells (Supplementary Fig. 4e). These findings support the proposal that, upon ROS-induced replication slowdown, R-loops form as a result of head-on TRCs leading to fork reversal. In this scenario, MUS81 and PRIMPOL would be required for replication restart when fork reversal is prevented by the elimination of R-loops, but fork progression is still impaired by the transcription complex, which may remain on the DNA after R-loop removal. In support of this notion, S1 nuclease treatment of permeabilized cells revealed that single-stranded DNA (ssDNA) gaps,

generated during PRIMPOL-mediated replication restart in HU-treated cells[5], were present not only upon ZRANB3 depletion but also upon RNase H1 overexpression, and not when transcription was inhibited (Fig. 4k; Supplementary Fig. 4f).

## R-loop-dependent fork stalling is induced by partial inhibition of DNA synthesis with aphidicolin

Our data, presented thus far, suggest that a reduction in replication fork velocity causes R-loop-mediated fork stalling associated with fork reversal. To verify this model, we analyzed the phenotypic consequences of exposure of cells to aphidicolin (APH), an inhibitor of the replicative DNA polymerases α, δ, and ε, which causes replication slowdown at low concentrations (0.1–1 μM) and replication arrest at high concentrations (>1 μM)[32]. By DNA fiber assay, we found that treatment of cells with 0.2 μM APH resulted in a sister fork asymmetry phenotype, which was rescued by overexpression of RNase H1 or inhibition of transcription (Fig. 5a, b; Supplementary Fig. 5a, b), indicating R-loop-mediated fork stalling. RNase H1 overexpression or transcription inhibition also partially restored normal replication fork velocity in these cells (Fig. 5c; Supplementary Fig. 5c). Moreover, U2OS cells treated with 0.2 μM APH displayed an increased number of PLA foci between the elongating form of RNAPII and PCNA (Fig. 5d, e), suggesting an increased incidence of TRCs. Strikingly, depletion of BRCA2 induced extensive nascent DNA degradation upon prolonged exposure of cells to 0.2 μM APH but not 2 μM APH (Fig. 5f). Moreover, 2 μM APH was found to completely inhibit HU-induced fork degradation in BRCA2-depleted cells (Fig. 5f). Taken together, these results support our conclusion that R-loop-mediated TRCs associated with fork reversal can result as a consequence of a global slowdown of replication fork progression, and that fork reversal is not induced by replication arrest per se.

## Replication slowdown leads to accumulation of R-loop-dependent DNA damage in nascent G1 cells

Our QIBC analyses revealed that HU treatment or TIMELESS depletion induced the accumulation of RNH1(D210N)-GFP foci not only in S-phase cells but also to some extent in G2 cells (Fig. 3g, k), suggesting the presence of the sites of persistent R-loop-mediated fork stalling. It is well known that regions of underreplicated DNA impair chromosome segregation during mitosis, leading to micronucleation and accumulation of DNA damage in nascent G1 cells[33]. We, therefore, investigated whether these phenotypes are induced by prolonged exposure of cells to 50 μM HU. To assess R-loop dependence, experiments were carried out in the U2OS T-REx cells line with inducible expression wild-type RNase H1. To detect micronuclei resulting from defective chromosome segregation, cells were treated for 16 h with cytochalasin B, which prevents cytokinesis while leaving karyokinesis unaffected[34]. DNA damage was monitored through immunofluorescence staining of the DNA-damage marker 53BP1, which forms characteristic nuclear bodies at sites of DNA double-strand breaks (DSBs) in G1 cells. We found that a 16-h exposure of cells to 50 μM HU increased the formation of micronuclei and G1-specific 53BP1 nuclear bodies in an R-loop-dependent manner (Fig. 6a–d). Similar results were obtained with cells treated with 0.2 μM APH (Supplementary Fig. 6a–d). Moreover, increased formation of R-loop-dependent micronuclei was seen in TIMELESS-depleted cells as compared to mock-depleted cells (Fig. 6e). Importantly, TIMELESS depletion did not further exacerbate the micronucleation phenotype induced by 50 μM HU in U2OS cells (Fig. 6f). In addition, we found that the formation of micronuclei in response to HU treatment was attenuated by depletion of PRDX2 (Fig. 6g). Taken together, these data suggest that R-loop-mediated fork stalling induced by replication slowdown leads, if persistent, to DNA breakage during chromosome segregation, which induces DNA-damage response in nascent G1 cells.

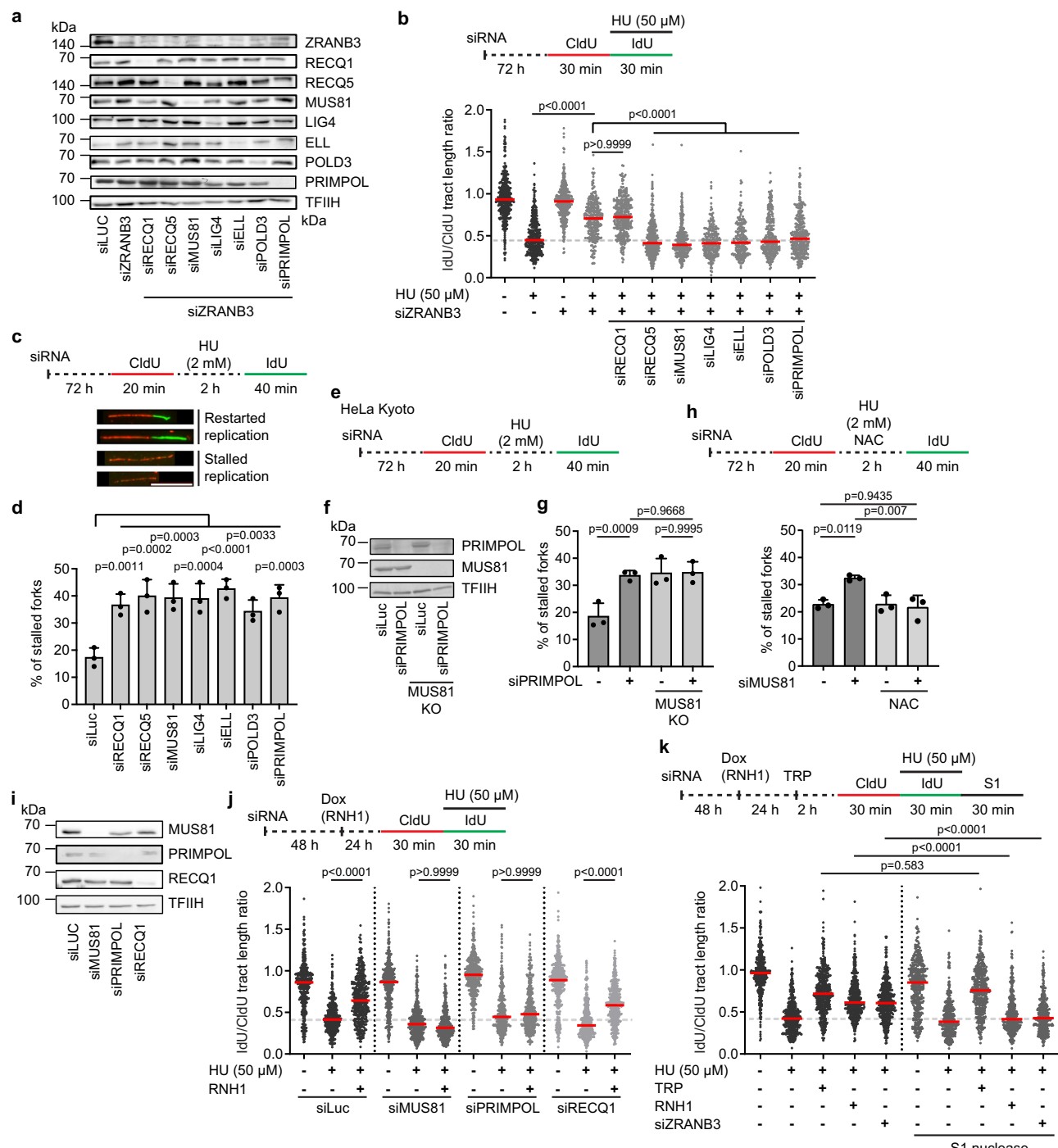

## Prolonged replication arrest due to deoxyribonucleotide shortage leads to R-loop-independent DNA breakage

Prolonged exposure of cells to high concentrations of HU leads to nucleus-wide exhaustion of RPA due to excessive amounts of ssDNA generated upon replication arrest, followed by massive DNA breakage in S-phase[4,5]. To distinguish whether this genome integrity breakdown is a consequence of ROS-induced replication slowdown or dNTP shortage, we treated U2OS cells with 4 mM HU for 4 h in the presence of NAC or exogenous nucleosides. QIBC analysis revealed that about 30% of HU-treated cells displayed elevated levels of chromatin-associated S4/8 phosphorylated-RPA2 (RPA2 pS4/8) (Fig. 7a–c), which is indicative of DSBs[35]. Cells with high levels of RPA loading also showed a strong signal of the DNA-damage marker γH2AX (Supplementary Fig. 7a, b). Notably, upon supplementation of cells with exogenous nucleosides, the HU-induced phosphorylation of RPA2 and H2AX were substantially diminished (Fig. 7a, b; Supplementary Fig. 7a, b). In contrast, the addition of NAC (Fig. 7a, b; Supplementary Fig. 7a, b), or the overexpression of RNase H1 (Fig. 7d, e; Supplementary Fig. 7c–e) failed to significantly affect these phenomena. These data suggest that DNA breakage induced by prolonged exposure of cells to a high concentration of HU is primarily the consequence of dNTP shortage. Correspondingly, the level of total RPA2 on chromatin in HU-treated cells could be profoundly reduced upon nucleoside supplementation, but it remained above the levels seen in untreated cells (Fig. 7c, f). As this level could be further decreased by a combination of nucleoside supplementation and RNase H1 overexpression (Fig. 7f), it appears that R-loops and dNTP depletion exert their respective negative effects on replication fork progression independently.

**Fig. 4 | MUS81-LIG4-PRIMPOL axis mediates replication restart following ROS-induced fork stalling. a** Western blot analysis of the extracts of U2OS cells transfected with indicated siRNAs. **b** ZRANB3 depletion restores replication fork progression in HU-treated U2OS cells in a manner dependent on PRIMPOL and the proteins required for restarting R-loop-stalled forks. *Top*: Experimental workflow. *Bottom*: Scatter plot of the values of IdU/CldU tract length ratio obtained from two independent experiments for indicated conditions ($n \geq 211$). **c, d** Restart of DNA synthesis following replication arrest by HU depends on the same set of proteins as in **b**. **c** Experimental workflow of replication restart assays and representative images of replication tracts corresponding to restarted and stalled forks, respectively. Scale bar, 10 μm. **d** Quantification of the replication fork stalling events in cells depleted of indicated proteins. **e–g** MUS81 and PRIMPOL act in the same replication restart pathway. **e** Experimental workflow of replication restart assays with wild-type and MUS81 knockout (KO) HeLa Kyoto cells. **f** Western blot analysis of the extracts of cells in **e** transfected with indicated siRNAs. **g** Quantification of replication fork stalling events for the indicated conditions. **h** HU-induced fork stalling in MUS81-depleted U2OS cells depends on ROS. *Top*: Experimental

workflow of replication restart assays. NAC, *N*-acetyl cysteine (5 mM). Bottom: Quantification of the replication fork stalling events for indicated conditions. **d, g, h** Data are presented as mean ± SD, $n = 3$; *p* values were calculated by one-way ANOVA followed by Tukey's test. **i** Western blot analysis of the extracts of U2OS T-REx [RNH1(WT)-GFP] cells transfected with indicated siRNAs. **j** Restoration of replication fork progression in HU-treated cells by RNase H1 overexpression depends on MUS81 and PRIMPOL, but not RECQ1. *Top*: Experimental workflow of DNA fiber assays with U2OS T-REx [RNH1(WT)-GFP] cells. Doxycycline (Dox; 1 ng/ml) was added to induce RNase H1 (RNH1) expression. *Bottom*: Scatter plot of the values of IdU/CldU tract length ratio obtained from three independent experiments for indicated conditions ($n \geq 406$). **k** Sensitivity of DNA replication tracts to S1 nuclease upon indicated conditions. *Top*: Experimental workflow of DNA fiber assays. TRP, triptolide (1 μM). *Bottom*: Scatter plot of the values of IdU/CldU tract length ratio obtained from three independent experiments for indicated conditions ($n \geq 417$). **b, j, k** Red horizontal lines indicate the median; *p* values were calculated by Kruskal-Wallis test followed by Dunn's multiple comparisons test. Source data are provided as a Source Data file.

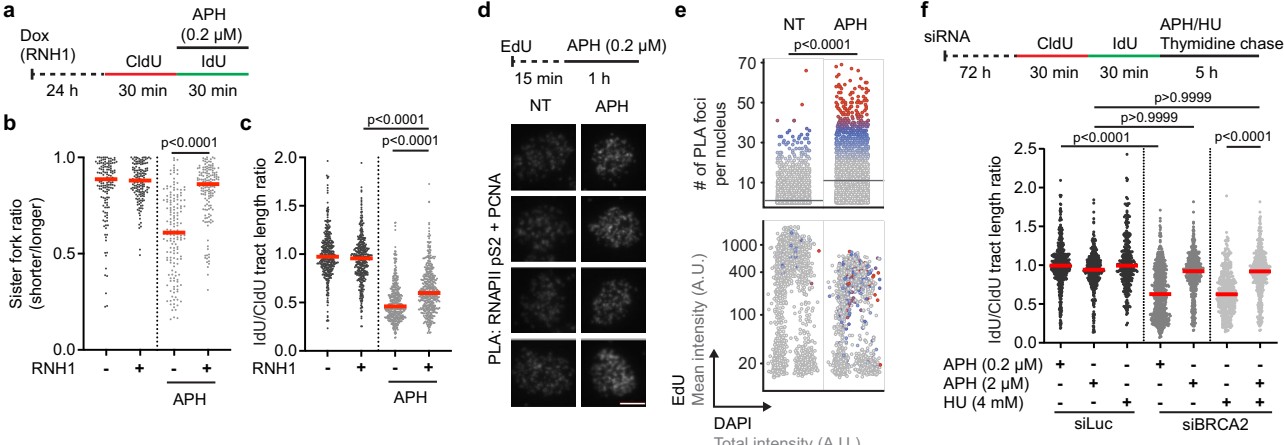

**Fig. 5 | R-loop-dependent fork stalling is induced by partial inhibition of DNA synthesis with aphidicolin. a–c** RNase H1 overexpression rescues replication fork stalling induced by a low dose of aphidicolin (APH). **a** Experimental workflow of DNA fiber assays with U2OS T-REx [RNH1(WT)-GFP] cells. Doxycycline (Dox; 1 ng/ml) was added to induce the expression of RNase H1 (RNH1). **b** Scatter plot of the values of sister fork ratio obtained from three independent experiments for indicated conditions ($n \geq 153$). **c** Scatter plot of the values of IdU/CldU tract length ratio obtained from three independent experiments for indicated conditions ($n \geq 402$). **d, e** Co-localization of PCNA and elongating form of RNAPII in S-phase nuclei of U2OS cells after 1 h treatment with 0.2 μM APH, as revealed by proximity ligation assay (PLA) and EdU-pulse labeling. **d** *Top*: Experimental workflow. EdU, 5-ethynyl-2′-deoxyuridine (10 μM). *Bottom*: Representative galleries of PLA signal in cell nuclei exported from ScanR analysis program. Scale bar, 10 μm. **e** *Top*: Scatter plot of the

number of PLA foci in individual cells for indicated conditions ($n \geq 2064$). Gray bars indicate the median; *p* values were calculated by Mann-Whitney test. *Bottom*: Scatter plot of total DAPI (x-axis) and mean EdU (y-axis) intensities in individual cells. Colors indicate the number of PLA foci and correspond to the upper plot. Representative plots from three independent experiments yielding similar results are shown. NT non-treated. **f** Low but not high doses of APH induce nascent DNA degradation in BRCA2-depleted U2OS cells. *Top*: Experimental workflow of DNA fiber assays. A thymidine (400 μM) chase was included for all conditions to stop IdU incorporation. *Bottom*: Scatter plot of the values of IdU/CldU tract length ratio obtained from two independent experiments for indicated conditions ($n \geq 332$). **b, c, f** Red horizontal lines indicate the median; *p* values were calculated by Kruskal-Wallis test followed by Dunn's multiple comparisons test. Source data are provided as a Source Data file.

# Discussion

R-loops generated due to impaired pre-mRNA processing or DNA topoisomerase 1 deficiency have been shown to induce stalling of DNA replication forks leading to genomic instability[36,37]. However, R-loops can also form as a result of transcription-replication encounters with essentially the same outcome[15,38]. Here, we present several lines of evidence suggesting that the global replication slowdown caused by ROS-induced dissociation of the TIMELESS-TIPIN complex from the replisome increases the incidence of transcription-replication interference which leads to R-loop formation and subsequent fork reversal, a DNA transaction that halts fork progression. Interestingly, R-loop-mediated TRCs also occur at late replicating common fragile sites (CFSs) upon partial inhibition of the replicative DNA polymerases with low doses of APH, causing CFS instability[39]. R-loop formation also underlies APH-induced mitotic DNA synthesis (MiDAS), a process that

serves to complete DNA replication at CFSs in early mitosis and thus prevents chromosome missegregation[13,17,40]. Similarly to what was observed for ROS-generating drugs in this study, exposure of cells to low APH concentrations also induced fork reversal, sister fork asymmetry and co-localization of PCNA and RNAPII[11,41,42]. In addition, our experiments showed that replication fork stalling induced by low doses of APH depends on transcription and R-loop formation (Fig. 5a–c; Supplementary Fig. 5a–c). Thus, it is tempting to speculate that R-loop-mediated TRCs and subsequent fork reversal generally occur if transcription complexes encounter a slowly-moving replisome. Of note, a recent study has shown that blocking lagging-strand priming using a POLα inhibitor slowed both leading- and lagging-strand synthesis due in part to fork reversal[43]. It will be interesting to see whether fork reversal induced by POLα inhibition is also a consequence of transcription-replication interference associated with R-loop formation.

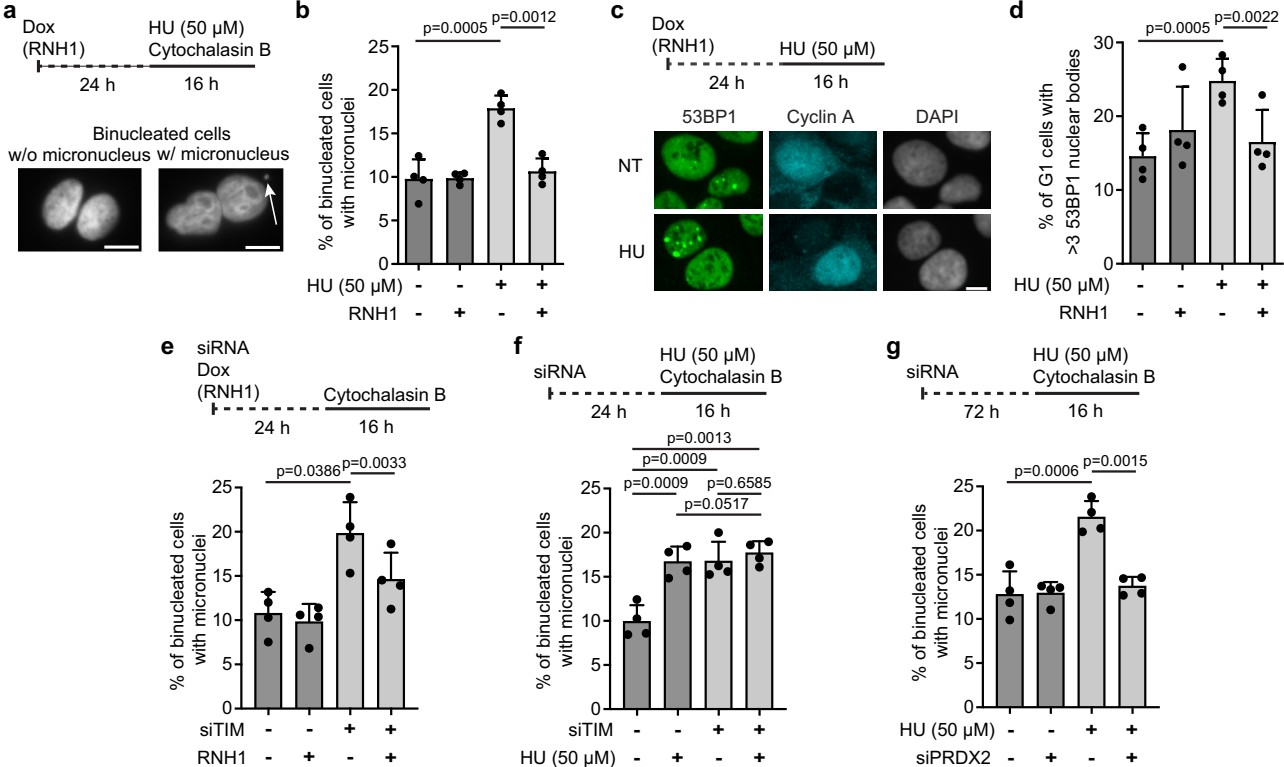

**Fig. 6 | Replication slowdown leads to R-loop-dependent micronucleation and accumulation of 53BP1 nuclear bodies in nascent G1 cells. a, b** HU induces R-loop-dependent micronucleation. **a** *Top*: Experimental workflow. U2OS T-REx [RNH1(WT)-GFP] cells were treated with 50 μM HU for 16 h. RNase H1 expression was induced by doxycycline (Dox; 1 ng/ml) 24 h before HU addition. Cytochalasin B (2 μg/ml; 16 h) was added to block cells in cytokinesis. *Bottom*: Representative images of binucleated cells without or with (white arrow) a micronucleus. Scale bar, 10 μm. **b** Quantification of the frequency of micronuclei for indicated conditions. **c, d** R-loop-dependent formation of G1-specific 53BP1 nuclear bodies in U2OS T-REx [RNH1(WT)-GFP] cells treated with 50 μM HU for 16 h. **c**, *Top*: Experimental workflow. *Bottom*: Representative images of DAPI, cyclin A and 53BP1 channels. Scale bar, 10 μm. **d** Percentages of cyclin A-negative cells (G1) with > 3 53BP1 nuclear bodies for indicated conditions. **e** TIMELESS depletion induces R-loop-dependent micronucleation. Percentage of binucleated U2OS T-REx [RNH1(WT)-GFP] cells with

micronuclei is plotted for indicated conditions. siTIM or siLuc were transfected for 24 h. Cytochalasin B (2 μg/ml) was added for the last 16 h before fixation to block cells in cytokinesis. RNH1 expression was induced with doxycycline (1 ng/ml) 24 h before siRNA transfection. **f** TIMELESS depletion does not further exacerbate the micronucleation phenotype induced by 50 μM HU in U2OS cells. *Top*: Experimental workflow. *Bottom*: Quantification of the frequency of micronuclei for indicated conditions. **g** HU-induced micronucleation is suppressed by PRDX2 depletion. *Top*: Experimental workflow. *Bottom*: Quantification of the frequency of micronuclei for indicated conditions. **b, d, e, f, g** Data are presented as mean ± SD, *n* = 4; *p* values were calculated by one-way ANOVA followed by Tukey's test. **b, e–g** For each condition, at least 150 binucleated cells were examined for the presence of micronuclei in each experiment. **d** For each condition, at least 300 cyclin A-negative cells were analyzed in each experiment to determine the number of 53BP1 nuclear bodies. Source data are provided as a Source Data file.

We propose that, under normal conditions, head-on transcription-replication encounters rarely lead to interference because the replisome can efficiently bypass the oncoming transcription complexes (Fig. 7g, green path), as shown for the DNA replication apparatus of bacteriophage T4[44]. In case of head-on encounters between a transcription complex and an impaired replisome, this bypass ability might be compromised, favoring R-loop formation and subsequent fork reversal (Fig. 7g, yellow path). However, it is possible that head-on TRCs give rise to R-loop-mediated fork stalling also under unchallenged conditions, albeit with lower efficiency. In fact, in genes containing origins of replication within the gene body, R-loops were found to be enriched in the region between the promoter and the origin in untreated HeLa cells[15]. Moreover, we found that the depletion of MUS81 and other proteins involved in the resolution of R-loop-mediated TRCs, such as RECQ5 or SLX4, increased the frequency of reversed forks in untreated U2OS cells[13]. Nevertheless, given the presence of relatively high levels of ROS in U2OS cells[10], it will be interesting to investigate whether fork stalling induced by the depletion of these proteins is dependent on ROS.

We found that in the presence of NAC, cells treated with 500 μM HU did not show an elevated frequency of reversed forks suggesting that DNA replication arrest resulting from HU-induced exhaustion of

dNTP pools does not trigger replication fork reversal (Fig. 2k). Similarly, complete inhibition of DNA synthesis by high APH concentrations did not result in nascent DNA degradation in BRCA2-depleted cells, which can be used as a proxy for fork reversal activity[20] (Fig. 5f). Instead, our data suggest that persistent replication arrest due to dNTP shortage results in unprotected replication forks that undergo a massive R-loop-independent DNA breakage during S-phase (Fig. 7a–f)[5], which presumably leads to cell death (Fig. 7g, red path). In contrast, R-loop-dependent fork stalling does not appear to give rise to extensive DNA damage during S-phase (Fig. 7d–f; Supplementary Fig. 7c–e)[10], most likely due to fork stabilization by fork reversal. However, if TRCs are not resolved before the onset of mitosis, the persistent regions of underreplicated DNA can impair chromosome segregation, leading to DNA breakage during cell division[33], which can give rise to chromosomal rearrangements in the nascent G1 cells[22]. Our observation that prolonged exposure of cells to low concentrations of HU or APH increased the incidence of R-loop-dependent micronuclei and G1-specific 53BP1 nuclear bodies would appear to support this hypothesis (Fig. 6a–d; Supplementary Fig. 6a–d). It should also be noted that many recurrent chromosomal rearrangements found in human cancers have been found to map to sites of TRCs and transcription-dependent DNA breakage[22,45].

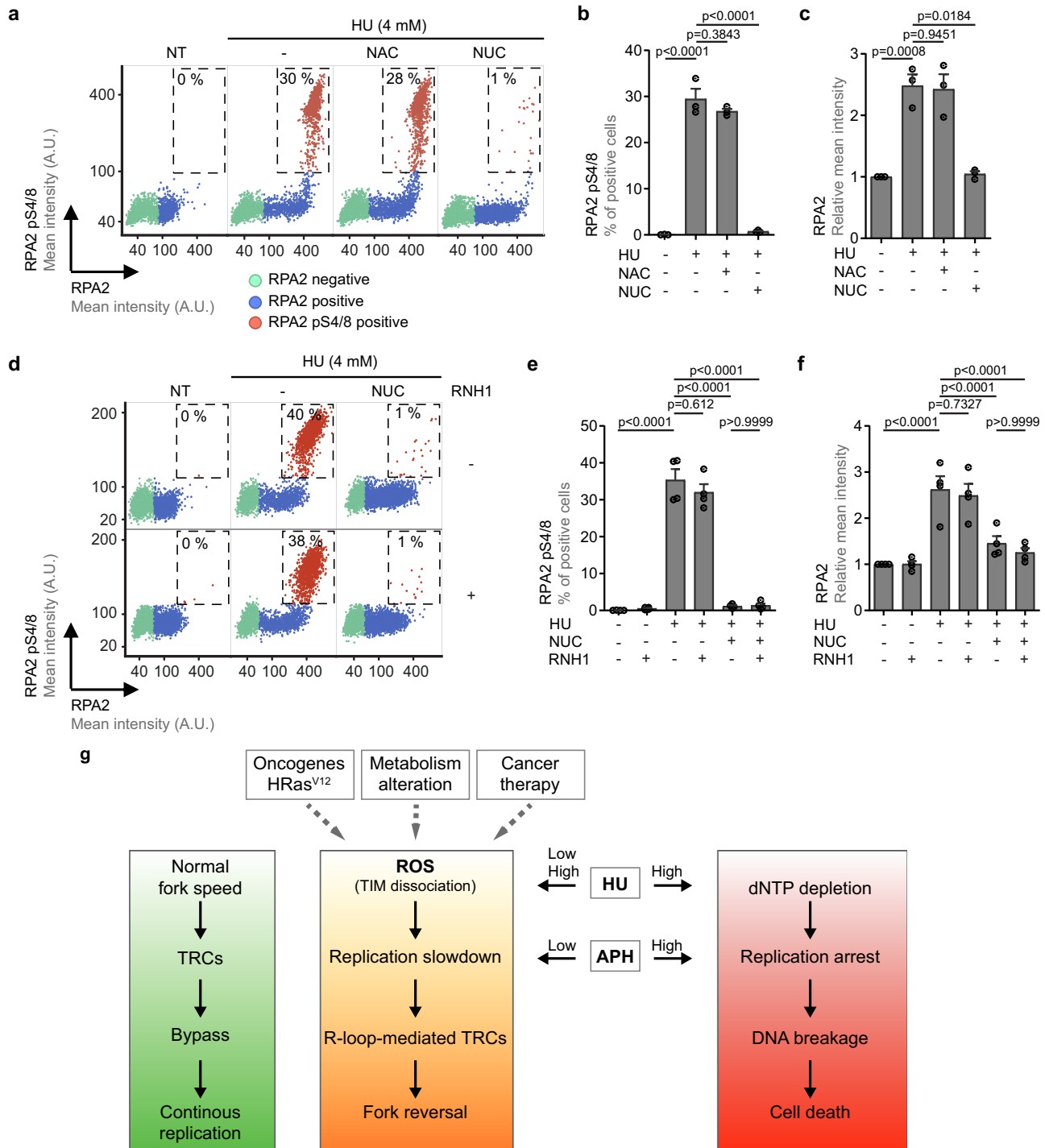

**Fig. 7 | Replication arrest due to dNTP shortage leads to R-loop-independent DNA breakage. a** Scatter plots showing the mean intensity of RPA2 (x-axis) versus the mean intensity of RPA2 pS4/8 (y-axis) in individual U2OS cells for indicated conditions, as measured by QIBC. Representative plots from three independent experiments yielding similar results are shown. At least 1000 cells were analyzed for each condition in each experiment. Cells were treated with 4 mM HU for 4 h in the absence or presence of *N*-acetyl cysteine (NAC; 5 mM) or exogenous nucleosides (NUC; 20 µM each). Individual cells are colored based on RPA and RPA2 pS4/8 intensities and classified as follows: RPA-negative (green), RPA-positive (blue) and RPA2 pS4/8-positive (red; dashed box). The numerical values inside the plots represent the percentage of RPA2 pS4/8-positive cells in the population. NT, non-treated; A.U., arbitrary units. **b** Plot of the percentage of RPA2 pS4/8-positive cells

from the QIBC analysis represented in (**a**). **c** Plot of the mean intensity of RPA2 from the QIBC analysis represented in (**a**). Data are normalized to the values obtained for NT. **d**–**f** The same as in (**a**–**c**) for U2OS T-REx [RNH1(WT)-GFP] cells. Representative plots from four independent experiments yielding similar results are shown. RNase H1 (RNH1) expression was induced by doxycycline (1 ng/ml) 24 h before the addition of HU. **g** A model of consequences of replication fork slowdown and replication arrest. ROS levels are frequently elevated in cancer cells by oncogenes such as HRasV12 or as a consequence of alterations in the cellular metabolism. ROS can be also generated during cancer therapy by ionizing radiation or chemotherapeutics such as mitomycin C or cisplatin. **b**, **c**, **e**, **f** Data are presented as mean ± SEM; *p* values were calculated by one-way ANOVA followed by Tukey's test. Source data are provided as a Source Data file.

Understanding the phenotypic consequences of oxidative stress is of substantial importance because ROS levels are frequently elevated in cancer cells by activated oncogenes such as HRasV12[46], and as a consequence of altered cellular metabolism or hypoxia[20,47]. ROS are also generated during radiotherapy and are a byproduct of the metabolism of chemotherapeutics such as mitomycin C and cisplatin[48–50]. The finding that oxidative stress triggers transcription-dependent replication stress, which appears to be a major source of chromosomal rearrangements in human cancers[45], helps explain how elevated ROS levels might contribute to tumor formation and progression. It will be interesting to learn whether inhibitors of the MUS81-LIG4-PRIMPOL pathway might potentiate the cytotoxic effects of the above chemotherapeutics.

## Methods

### Cell culture

U2OS (ATCC HTB-96), HeLa Kyoto (ATCC CVCL 1922) and RPE1 (ATCC CRL-4000) cells were grown in Dulbecco's Modified Eagle Medium (DMEM; Thermo Fisher Scientific), supplemented with 10% fetal calf serum (FCS; Thermo Fisher Scientific) and streptomycin/penicillin (100 U/ml), at 37 °C in a humidified incubator containing 5% CO2. MUS81 knockout HeLa Kyoto cells were previously described[13]. U2OS T-REx cell lines carrying pAIO-based vectors for conditional expression of GFP-tagged wild-type RNase H1 or RNase H1 D210N (RNase H1 ORF fused C-terminally to GFP) were cultivated in DMEM supplemented with 10% FBS (Tet-free approved), streptomycin/penicillin (100 U/ml), 50 µg/ml hygromycin B and 1 µg/ml puromycin[24]. RNase H1-GFP expression was induced by the addition of doxycycline to a concentration of 1 ng/ml.

### Small-interfering RNA transfection

Transfections of siRNAs (a final concentration of 40 nM) were done using Lipofectamine RNAiMAX (Invitrogen) according to the manufacturer's instructions (reverse transfection protocol). 24 h after siRNA transfection, the medium was exchanged with fresh medium. Experiments were performed 72 h after siRNA transfection. TIMELESS depletion was achieved by transfection of 5 or 10 nM siRNA for 24 h. The majority of siRNA oligonucleotides used in this study were purchased from Microsynth AG. The sequences of the sense strand of these siRNA duplexes are:

5′-CGUACGCGGAAUACUUCGA-3′ (siLUC siRNA); 5′-CAGGACACA AUUACAACUAAA-3′ (BRCA2 siRNA); 5′-CAACAAGGCACCAGGGAAA-3′ (POLD3 siRNA); 5′-GCUAGAUGGUGAACGUAUG (LIG4 siRNA_1); 5′-AA GCCAGACAAAAGAGGUGAA-3′ (LIG4 siRNA_2); 5′-GAGGAAAGCUGG ACAUCGA-3′ (PRIMPOL siRNA); 5′-AAUGAGACUUACGUG UUAAAAUT-3′ (RIF1 siRNA); 5′-AACAAGTGAATTGCCGCAGAA-3′ (HLTF siRNA); 5′-G CAAGGAGAUUUACUCGAA-3′ (RECQ1 siRNA); 5′-CAGGUUUGUCGC CCAUUGGAA-3′ (RECQ5 siRNA); 5′-CAGCCCUGGUGGAUCGAUA-3′ (MUS81 siRNA); and ZRANB3 siRNA was purchased from Dharmacon (84083; D-010025-03-0005), ELL siRNA was purchased from Santa Cruz Biotechnology (sc-38041) and TIMELESS (s17053) and PRDX2 (s13960) siRNAs were purchased from Thermo Fisher Scientific.

### DNA fiber spreading assay

Cells were sequentially pulse-labeled with 30 µM 5-chloro-2-deoxyuridine (CldU; Sigma-Aldrich) and 250 µM 5-iodo-2-deoxyuridine (IdU; Sigma-Aldrich) for 30 min each. Where indicated, thymidine (400 µM) chase was used. After labeling, cells were washed three times with PBS, quickly trypsinized and resuspended in PBS to a concentration of 250,000 cells per ml. 2.5 µl of this cell suspension were mixed with 7.5 µl of lysis buffer [200 mM Tris-HCl (pH 7.5), 50 mM EDTA, 0.5% (w/v) SDS] on a glass slide by gently stirring with a pipette tip. After 9-min incubation at room temperature (RT), the slides were tilted at 30-40 degrees, and the drops were allowed to run down the slides slowly. The DNA spreads were air-dried and fixed in

methanol/acetic acid (3:1) at 4 °C overnight. DNA fibers were denatured with 2.5 M HCl for 1 h at RT, washed four times with PBS and blocked with 2.5% BSA and 0.1% Tween-20 in PBS for 40 min. After blocking, slides were incubated for 2 h in the dark at RT with rat monoclonal anti-BrdU antibody (ab6326, Abcam; 1:500 dilution) to detect CldU and mouse monoclonal anti-BrdU antibody (347580, BD Biosciences; 1:100 dilution) to detect IdU. Slides were then washed four times with PBST (PBS supplemented with 0.2% Tween-20) and incubated with secondary antibodies, donkey anti-rat Cy3 (712-166-153, Jackson ImmunoResearch; 1:150 dilution) and goat anti-mouse Alexa 488 (A110334, Thermo Fisher Scientific; 1:300 dilution), for 1 h in the dark at RT. After washing four times with PBST, the slides were air-dried in the dark for 40 min at RT and mounted with ProLong Gold antifade mounting medium (Thermo Fisher Scientific, 25 µl per coverslip 24×50 mm). Images were acquired with a Leica DM6000 upright fluorescent microscope (63x/1.40 Oil immersion). CldU and IdU tract lengths were measured by using the segmented line tool of ImageJ. Data from three independent experiments were combined.

### S1 nuclease assay

Cells were pulse-labeled with CldU and IdU as described above. After incubation with IdU, cells were washed with PBS and pre-extracted with CSK buffer [25 mM HEPES (pH 7.7), 50 mM NaCl, 1 mM EDTA, 3 mM MgCl₂, 300 mM sucrose, 0.5% Triton X-100] for 10 min at RT. Cells were then carefully washed with PBS, and with S1 buffer [30 mM Sodium Acetate (pH 4.6), 10 mM Zinc Acetate, 5% glycerol, 50 mM NaCl] and incubated with or without S1 nuclease (Thermo Fisher Scientific, 20 U/ml) in S1 buffer for 30 min in 37 °C. After incubation, S1 buffer was changed for 0.1% BSA in PBS and cells were scraped into an Eppendorf tube. The cell suspension was centrifuged at $4600 \times g$ for 10 min and the supernatant was discarded. Cells were resuspended in PBS to a concentration of ~1500 cells per µl. 2.5 µl of this cell suspension was mixed with 7.5 µl of lysis buffer [200 mM Tris-HCl (pH 7.5), 50 mM EDTA, 0.5% (w/v) SDS] on a glass slide by gently stirring with a pipette tip. After 4-min incubation at RT, the slides were tilted at 30-40 degrees, and the drops were allowed to run down the slides slowly. The rest of the procedure is identical to the DNA fiber spreading assay described above.

### Immunofluorescence staining

Cells grown on autoclaved coverslips were transfected with siRNA and/or treated with drugs as indicated. After the treatment, cells were washed with PBS and pre-extracted for 5 min with ice-cold CSK buffer. Then the cells were washed with CSK buffer without Triton X-100 and fixed with 4% formaldehyde (Sigma-Aldrich) for 15 min at RT. After three washes with PBS, fixed cells were incubated in PBS containing 0.2% Triton X-100 for 10 min at RT. Cells were then washed with PBS and blocked in 3% BSA/PBS solution for 30 min. Coverslips were then incubated with appropriate primary antibodies diluted in 3% BSA/PBS for 90 min at RT. Then, the coverslips were washed three times with PBS and incubated for 60 min at RT with secondary antibodies diluted in 3% BSA/PBS. After three washes with PBS, coverslips were incubated with 1 µg/ml DAPI/PBS for 10 min in dark at RT. The coverslips were then washed twice with PBS and mounted with Fluoromount-G mounting medium (Thermo Fisher Scientific). For the sequential labeling with two mouse antibodies (RPA2 and γH2AX), the coverslips were initially incubated with the primary anti-γH2AX antibody for 90 min at RT, washed three times with PBS, and then incubated with secondary Alexa Fluor 488 Goat Anti-mouse antibody for 30 min RT. Coverslips were then washed three times with PBS and blocked with 3% BSA/PBS solution for 30 min, followed by incubation with the primary anti-RPA2 antibody for 90 min at RT and secondary Alexa Fluor 647 Goat Anti-mouse antibody for 30 min. The primary antibodies used for the immunofluorescence staining: RPA2 (9H8) mouse monoclonal (ab2175, Abcam; 1:400 dilution), γH2AX (S139) mouse monoclonal

(05-636, Merck-Millipore; 1:400 dilution), RPA2 pS4/8 rabbit polyclonal (A300-245A, Bethyl Laboratories; 1:400 dilution), PCNA (PC10) mouse monoclonal (sc56, Santa Cruz Biotechnology; 1:500 dilution), 53BP1 rabbit polyclonal (sc22760, Santa Cruz Biotechnology; 1:400 dilution), Cyclin A (B-8) mouse monoclonal (sc-271682, Santa Cruz Biotechnology, 1:200 dilution). The secondary antibodies used for immunofluorescence staining: Alexa Fluor 488 Goat Anti-Rabbit IgG (A110334, Thermo Fisher Scientific; 1:400 dilution), Alexa Fluor 594 Goat Anti-Rabbit IgG (A11037, Thermo Fisher Scientific; 1:400 dilution), Alexa Fluor 488 Goat Anti-Mouse IgG (A11001, Thermo Fisher Scientific; 1:400 dilution), Alexa Fluor 594 Goat Anti-Mouse IgG (A11005, Thermo Fisher Scientific; 1:400 dilution), Alexa Fluor 647 Goat Anti-Mouse IgG (A21235, Thermo Fisher Scientific; 1:400 dilution).

### Quantitative image-based cytometry

Images were acquired in an unbiased fashion with an Olympus IX81 or IX83 automated inverted microscope equipped with a ScanR imaging platform. After the acquisition, the images were analyzed with automated Olympus ScanR analysis 3.0.1. software. DAPI signal was used for generating a mask that identified each individual nucleus as an individual object. This mask was then applied to quantify pixel intensities in the different channels for each individual cell/object. After quantification, the quantified values for each cell (mean and total intensities, area, number of sub-objects) were extracted and exported to the Tibco Spotfire software. Tibco Spotfire was used to visualize key features of replication stress and DNA-damage signaling and quantify percentages and average values in cell populations. Spotfire filtered data were then used to generate plots using GraphPad Prism 9 software (version 9.4.1). XY scatter plots in the figures show a representative experiment out of three independent experiments. Percentages of the population are calculated as a mean value from three experiments. RPA2 and γH2AX intensities are plotted as a mean value from three independent experiments with intensities normalized to values obtained for non-treated (NT) conditions.

### In situ proximity ligation assay

U2OS cells were grown on autoclaved coverslips and treated with 20 μM EdU for 15 min, washed with PBS, and treated with drugs for 1 h as indicated in Figure legends. After treatment, cells were washed twice with PBS and pre-extracted for 10 min with ice-cold CSK buffer supplemented with a protease inhibitor cocktail (Complete, EDTA-free; Sigma-Aldrich). Then, the cells were washed once with CSK buffer without Triton-X100 and fixed with 4% formaldehyde for 15 min at RT. After two washes with PBS, fixed cells were additionally fixed with −20 °C methanol and left in the freezer for 20 min. Cells were then washed three times with PBS and blocked in 3% BSA/PBS solution for 40 min. Coverslips were then incubated with Click-IT EdU staining mixture with Alexa-Azide 647 nm (A10277, Thermo Fisher Scientific) for 30 min at RT. After washing twice with PBS, coverslips were incubated O/N at 4 °C with appropriate primary antibodies diluted in 3% BSA/PBS. Both rabbit polyclonal anti-PCNA (ab18197, Abcam) and mouse monoclonal anti-RNAPII, H5 (920204, BioLegend) were used in a dilution of 1:1000. The following day, coverslips were washed three times with PBS and proximity ligation assay (PLA) was performed using Duolink PLA technology (Sigma-Aldrich) according to the manufacturer's instructions. Images were acquired with an Olympus IX83 microscope and analyzed using the Olympus ScanR analysis program. Results of the analysis were exported into Tibco Spotfire and visualized.

### Electron microscopy

U2OS T-REx [RNH1(WT)-GFP] cells grown to 70% confluency were treated with 500 μM HU for 1 h. Where indicated, cells were pretreated for 24 h with 1 ng/ml doxycycline to induce RNase H1 overexpression, or 2 h with 100 μM DRB or co-treated with NAC. Cells were subsequently harvested by trypsinization and crosslinked twice by addition of 4,5′,8-trimethylpsoralen (final concentration 10 μg/ml), followed by UV irradiation at 365 nm for 3 min (UV Stratalinker 1800; Agilent Technologies). Cells were then lysed for 10 min on ice in lysis buffer [40 mM Tris-Cl (pH 7.5), 1.28 M sucrose, 20 mM MgCl₂, 4% Triton X-100]. Nuclei were collected by centrifugation at 250xg for 15 min, briefly washed with lysis buffer, and subsequently digested with digestion buffer [800 mM guanidine-HCl, 30 mM Tris-HCl (pH 8.0), 30 mM EDTA, 5% Tween-20, 0.5% Triton X-100, and 1 mg/ml proteinase K] at 50 °C for 2 h. Genomic DNA was extracted from lysates by 24:1 chloroform: isoamylalcohol phase separation at 8000 rpm for 20 min at 4 °C and precipitated from the upper aqueous phase with one volume of isopropanol followed by centrifugation at $7000 \times g$ for 10 min at 4 °C. The genomic DNA pellets were washed with 70% ethanol, briefly air-dried, and dissolved in TE buffer by gentle rotation overnight. 6 μg of the obtained genomic DNA were digested with 120 units of PvuII-HF for 5 h at 37 °C with the addition of 250 μg/ml RNase A during the last 2 h of this incubation. The digested DNA was purified using the Thermo Fisher Silica Bead Gel Extraction kit according to the manufacturer's instructions. For analysis by electron microscopy, the digested DNA was mixed with benzyldimethylalkylammonium chloride (BAC), spread on a water surface, and loaded onto carbon-coated 400-mesh magnetic nickel grids. The DNA-loaded grids were coated with 13 nm of platinum by platinum-carbon rotary shadowing and imaged automatically using a Talos 120 transmission electron microscope (FEI; LaB6 filament, high tension ≤120 kV) with a bottom-mounted CMOS camera BM-Ceta (4000x4000pixel) and the MAPS software package (Thermo Fisher Scientific, Eindhoven, The Netherlands). Samples were annotated for replication intermediates using the MAPS Viewer software, overlapping images for annotated regions were stitched together using the automated pipeline ForkStitcher and final images were analyzed using ImageJ.

### Preparation of cell extracts and western blot analysis

Cells were washed with PBS and scraped into Eppendorf tubes in a lysis buffer containing 10 mM Tris-HCl (pH 7.5), 1% SDS and 1 mM EDTA. Cell lysates were sonicated three times for 15 s and boiled at 95 °C for 10 min. The solutions were clarified by centrifugation at 18,000xg for 10 min and the supernatant was transferred in a new Eppendorf tube. The protein concentration was determined by bicinchoninic acid assay. Cell lysates were then supplemented with DTT and Bromophenol blue. Samples containing 30–60 μg of total protein were loaded onto 8% or 10% SDS-PAGE gels. After electrophoresis, separated proteins were transferred from the gel onto a PVDF membrane in a wet-transfer apparatus (Bio-rad) with buffer containing 25 mM Tris, 19.2 mM glycine and 10% methanol, run overnight at 30 V and 4 °C. The membrane was blocked with 5% milk in TBS-T [20 mM Tris-HCl (pH 7.4), 150 mM NaCl, 0.1% Tween-20] for 30 min. Afterward, the membranes were incubated with the primary antibodies in 3% milk/TBS-T at 4 °C overnight. The membranes were then washed 3 times in TBS-T and incubated with appropriate horseradish peroxidase-coupled (HRP) secondary antibody in TBS-T for 60 min at RT. Afterward, the membranes were washed three times with TBS-T and protein bands were detected by luminol-based reaction using a chemiluminescence reagent (Pierce). For detection of large proteins such as BRCA2, cells were lysed in RIPA lysis buffer [50 mM Tris-HCl (pH7.4), 200 mM NaCl, 1% NP-40, 0.1% sodium deoxycholate, 1 mM EDTA] supplemented with protease inhibitor cocktail (Complete, EDTA-free; Sigma-Aldrich) on ice for 10 min. Protein concentration in clarified cell lysates was measured by Pierce 660 Protein assay reagent (Thermo Fisher Scientific). The primary antibodies used for western blotting: MUS81 (MTA30 2G103) mouse monoclonal (sc-53382, Santa Cruz Biotechnology; 1:500 dilution), TFIIH p89 (S-19) rabbit polyclonal (sc-293, Santa Cruz Biotechnology; 1:1000 dilution), RECQ5 rabbit polyclonal (Janscak lab; 1:1000 dilution), RECQ1 rabbit polyclonal (NB100-182, Novus

Biological; 1:1000 dilution), LIG4 (D-8) mouse monoclonal (sc-271299, Santa Cruz Biotechnology; 1:500 dilution), BRCA2 (Ab-1) mouse monoclonal (OP-95, EMD Millipore; 1:200 dilution), RNASE H1 (A-9) mouse monoclonal (sc-365783, Santa Cruz Biotechnology, 1:500 dilution), POLD3 (M01, clone 3E2) mouse monoclonal (H00010714-M01, Abnova; 1:1000 dilution), ZRANB3 rabbit polyclonal (23111-1-AP, Proteintech; 1:1000 dilution), ELL (B-4) mouse monoclonal (sc-398959, Santa Cruz Biotechnology; 1:500 dilution), HLTF mouse monoclonal (sc-398357, Santa Cruz Biotechnology; 1:500 dilution), Lamin B2 (gt144) mouse monoclonal (gtx628803, Genetex, 1:1000 dilution), TIMELESS (EPR5275) rabbit monoclonal (ab109512, Abcam, 1:500 dilution), PRDX2 mouse monoclonal (sc-515428, Santa Cruz Biotechnology; 1:500 dilution); PRIMPOL rat antibody was kindly provided by Dr. Juan Mendez[51]. The secondary antibodies used for western blotting: goat anti-rabbit IgG-HRP (A0545, Sigma-Aldrich; 1:5000 dilution), goat anti-mouse IgG-HRP (A4416, Sigma-Aldrich; 1:5000 dilution), goat anti-rat IgG-HRP (SC2006, Santa Cruz Biotechnology; 1:5000 dilution)

### Micronucleus assay
U2OS T-REx [RNH1(WT)-GFP] cells were transfected with siRNA as described above. After 24 h, cells were trypsinized and seeded on coverslips. RNase H1 expression was induced by the addition of doxycycline (1 ng/ml). After 24 h, cells were treated with cytochalasin B (2 µg/ml) and hydroxyurea or aphidicolin for 16 h. Cells were then fixed with 4% formaldehyde in PBS for 15 min at RT and subsequently permeabilized in PBS containing 0.2% Triton X-100 for 10 min at RT. Coverslips were incubated with 1 µg/ml DAPI/PBS for 10 min in dark at RT, washed with PBS, and mounted with Fluoromount-G antifade reagent. Images were acquired with a Leica DM6000 upright fluorescent microscope (40x dry objective). The percentage of binucleated cells containing micronuclei was determined. At least 150 binucleated cells were scored per condition in each experiment.

### DNA-RNA immunoprecipitation followed by quantitative real-time PCR
DRIP experiments were carried out as described[26], with a few minor modifications for U2OS cells. Briefly, nucleic acids isolated from cells using conventional phenol-chloroform extraction were digested with restriction enzymes including BamHI, BsrGI, EcoRI, HindIII and XhoI (20 U each, Thermo Fisher Scientific) for 36 h at 37 °C, either with or without the addition of RNaseH (40 U, New England Biolabs). Digested nucleic acids were cleaned by phenol-chloroform extraction and then resuspended in nuclease-free water (Sigma-Aldrich). For RNA:DNA hybrid immunoprecipitation, 2.5 µg of nucleic acids were incubated with 6.5 µg of mouse monoclonal anti-RNA:DNA hybrid (S9.6) antibody (ENH001, Kerafast) overnight in binding buffer [10 mM NaH$_2$PO$_4$/Na$_2$HPO$_4$ (pH 7.0), 140 mM NaCl, 0.05% Triton X-100] at 4 °C. 5% of DNA was removed as input. The following day, 25 µl of Pierce™ protein A/G magnetic beads (Thermo Fisher Scientific) were added and the mixture was incubated for 2 h at 4 °C. Beads were washed thrice with the binding buffer and incubated with elution buffer [50 mM Tris (pH 8), 10 mM EDTA, 0.5% SDS, 1 mg/ml proteinase K] for 45 min at 55 °C. Eluted DNA was purified by phenol:chloroform extraction, followed by ethanol precipitation and subjected to quantitative real-time PCR (qPCR) analysis using iQ SYBR Green Supermix reagent (Bio-Rad) and the gene-specific primers described below on a CFX96 Real-Time PCR detection system (Bio-Rad). All the data were analyzed using the $2^{-\Delta\Delta CT}$ method[52]. The percentage of DNA in the immunoprecipitates compared to the input DNA was calculated and plotted using Prism8 (GraphPad Software). The sequences of the primers used for qPCR are:

5′-CCGGTGAGAAGCGCAGTCGG-3′ (APOE-Forward); 5′-CCCAAGC CCGACCCCGAGTA-3′ (APOE-Reverse); 5′-GGCTGCTCAGGAGAGCTAG A-3′ (BTBD19-Forward); 5′-ACCAGACTGTGACCCCAAAG-3′ (BTBD19-Reverse); 5′-AATGTGGCATTTCCTTCTCG-3′ (RPL13A-Forward); 5′-CCA ATTCGGCCAAGACTCTA-3′ (RPL13A-Reverse); 5′-GGCTCAGAGGAGG GCTCTTT-3′ (BCL2-Forward); 5′-GTGCCTGTCCTCTTACTTCATTCT C-3′ (BCL2-Reverse); 5′-GCCTGTGAGTGAGTGCAGAA-3′ (GADD45A-Forward); 5′-CGACTCACCTTTCGGTCTTC-3′ (GADD45A-Reverse); 5′-T GCCAGGAAGCCAAATGAGT-3′ (SNRPN-Forward); 5′-TCCCTCTTGGC AACATCCA-3′ (SNRPN-Reverse).

### Statistical analysis
Statistical analysis was performed using GraphPad Prism 9 software (version 9.4.1). Details of how data are presented, including definition of center (mean or median) and error bars, as well as the details of statistical tests for each experiment, including the type of statistical tests used and the number of repeats, can be found in the figure legends. Statistical test results, presented as $p$ values, are shown in the figures. Statistical differences in scatter plots of data from DNA fiber assays and QIBC analyses of PLA or RNH1(D210N)-GFP foci were determined by a two-tailed non-parametric Kruskal-Wallis test followed by a Dunn's multiple comparisons test. When only two conditions are compared, a two-tailed non-parametric Mann-Whitney test was used. Statistical differences for other grouped analyses, i.e., micronuclei, mean intensities of RPA2 and γH2AX or frequency of fork reversal (EM), were assessed by a repeated-measures one-way ANOVA followed by a Tukey's multiple comparisons test. All statistical test results are listed in the Source Data file Summary Statistics.

### Reporting summary
Further information on research design is available in the Nature Portfolio Reporting Summary linked to this article.

## Data availability
The authors declare that the data supporting the findings of this study are available within the paper and its Supplementary Information files. Supplementary Figs. 1–7 are provided within the paper. All other data that support the findings of this study are available from the authors upon reasonable request. Source data are provided with this paper.

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

## Acknowledgements

We thank Josef Jiricny for comments on the manuscript, Juan Mendez for providing PRIMPOL antibody and Christiane König for technical assistance. We also thank the UZH Center for Microscopy and Image Analysis and the Light Microscopy Core Facility of IMG (MEYS: LM2018129, CZ.02.1.01/0.0/0.0/18_046/0016045; RVO: 68378050-KAV-NPUI) for support. This work was supported by grants from the Swiss National Science Foundation (310030_184716), the Czech Science Foundation (22-08294 S), the Swiss Cancer League (KFS-5484-02-2022) and Foundation for Research in Science and the Humanities at the University of Zurich. M.L. was supported by the Swiss National

Science Foundation (310030_189206). H.S. was supported by grants from Forschungskredit UZH and Krebsliga Zurich. J.D. was supported by the Czech Science Foundation (21-22593X). R.K. and K.S. were co-investigators on the Children with Cancer (CwC) UK grant [PGTaSFA\100027].

## Author contributions

P.J. and M.A designed the study; M.A. performed most of the DNA fiber experiments, PLA assays and DNA-damage assays; H.S. and M.L. designed the EM experiments, fully performed by H.S.; B.B., R.K., Z.N., A.O., K.S., S.M. and J.D. performed the R-loop assays; B.B., N.C. and S.R. performed a part of the DNA replication restart experiments; P.J. supervised the project; P.J. and M.A. wrote the paper with contributions from M.L. and H.S.

## Competing interests

The authors declare no competing interests.
