## [Peer Review File · Nature Communications]

REVIEWER COMMENTS

Reviewer #1 (Remarks to the Author):

Previous studies suggested that low HU induces ROS to slowdown replication forks independently of dNTP depletion. In this study, the authors revisited the issue and suggested that ROS slows down replication forks by inducing co-transcriptional R-loops. They also present some data suggesting that several DNA repair/replication proteins are involved in the response to low HU. Finally, they show that R-loops are not responsible for the replication stress/DSBs induced by high HU. I really wanted to like this study, but I was quite disappointed as I read through the manuscript and figures. There are many issues that are not tested or convincingly addressed (see below). One major issue is that the model of this study is opposite from previous studies. Previous studies have suggested that replication fork reversal is a mechanism to slowdown forks. However, this study tries to argue that fork slowdown is the cause of fork reversal. This point is not well supported by the data. In addition, this study argues that replication slowdown increases R-loop-mediated TRCs, but direct evidence supporting this point is lacking. Although the model of this study is provocative, it is not convincing at this point. A much extended and reshaped study is needed for a paper in Nat Commun.

1. In the introduction, the authors should acknowledge that the slowdown of replication forks by co-transcriptional R-loops has been shown by James Manley's lab many years ago.

2. The authors should also acknowledge that ROS-induced co-transcriptional R-loops have been reported and studied (Teng et al. Nat Commun 2018).

3. In Fig. 1e, it is not entirely clear whether nascent DNA degradation is the cause of sister fork asymmetry. MRE11 may cause fork collapse and contribute to this asymmetry.

4. The data in ED Fig. 1d and 1e are inconsistent. Exogenous nucleosides did not rescue sister fork asymmetry at all in 500 uM HU, but partially rescued fork slowdown. Does this mean that fork slowdown and sister fork asymmetry are at least partially independent phenotypes? The effects of nucleosides on "nascent DNA degradation" is not tested in high HU. Overall, the relationships among nascent DNA degradation, fork slowing, and sister fork asymmetry are not worked out convincingly.

5. The authors heavily depend on NAC and Conoidin A for ROS-related conclusions, which is potentially problematic. These compounds could have unintended effects at the concentrations used. The use of low and high HU cannot "cleanly" separate ROS and dNTP depletion either. Can the authors confirm the results by depleting PRDX2 and RRM2 directly?

6. The Timeless depletion experiments in Fig. 1J-n should be done in the presence and absence of low HU. If HU-induced ROS release Timeless from the replisome, Timeless knockdown should bypass the requirement for low HU to slowdown forks. It is not clear whether the effects of Timeless on fork speed and sister fork symmetry are specific to Timeless. Some other replisome proteins should be tested as controls.

7. In Fig. 2a-d, the phenotypes of fork slowdown and sister fork asymmetry are inconsistent again. RNH1 only partially rescued fork slowdown, but fully rescued sister fork asymmetry. This raises further doubt on whether these phenotypes can be interpreted in the same way. It is also quite surprising that RNH1 fully rescued sister fork asymmetry in 2d, which means that dNTP depletion does not contribute to this phenotype at all. This should be retested by RRM2 depletion.

8. The results in Fig. 2e-g are very surprising. The previous model suggests that the ROS-induced release of Timeless from the replisome is the "cause" of fork slowing. However, the data here suggest that R-loops, rather than the loss of Timeless, are the real cause of fork slowdown. If this is the case, why should PRDX2 matter? A replisome-associated ROS sensor should not affect ROS-induced co-transcriptional R-loops ahead of forks.

9. The experiments in Fig. 2h-j should be done in low HU. If low HU (and ROS) is sufficient to slowdown forks and cause fork reversal, it should also be sufficient to induce nascent DNA degradation in BRCA2 knockdown cells. If the effects of ROS can be reversed by RNH1, these experiments should

work in low HU.

10. The EM experiments in Fig. 2k should also be done in low HU and in the absence of HU. If HU induces ROS to generate co-transcriptional R-loops, low HU should work identically as high HU. Furthermore, the inclusion of -HU controls is important for assessing how much RNH1/DRB/NAC suppress the HU effects.

11. In Fig. 3c-e, both high and low HU should be tested. If low HU is sufficient to induce ROS and R-loops, it should work identically to high HU.

12. Again, Fig. 3f should be done in low HU. If low HU is sufficient to induce R-loops, there is no reason to use 10 mM HU. Cells in early S phase may be particularly sensitive to the depletion of dNTPs.

13. As mentioned above, it is surprising that loss of Timeless induces R-loops because Timeless travels with replication forks but not the transcription complex. Presumably depletion of Timeless should reduce fork speed and replication-transcription collisions, which may reduce R-loops. The authors should check whether loss of Timeless affects transcription. They should also check whether the R-loops in Timeless knockdown cells are co-transcriptional.

14. Previous studies have shown that loss of fork reversal factors SMARCAL1/ZRANB3/HLTF increases fork speed in low HU or other conditions inducing fork reversal. If fork reversal is the consequence rather than the cause of fork slowdown as this study suggests, one would predict that loss of fork reversal should not alter fork slowdown, and the data from the previous studies would be difficult to explain. This is a key issue to be addressed by this study.

15. In Fig. 4, the effects of RNH1 on the cell cycle should be carefully monitored. A delay in cell cycle progression would affect all the phenotypes indirectly.

16. The data in Fig. 5 support the notion that fork reversal causes fork slowdown, but not the idea that fork reversal is a consequence of fork slowdown. The author switched the concepts when they moved from Fig. 4 to Fig. 5. This is not helpful to the overall conclusion of this study.

17. It is surprising that MUS81/LIG4/POLD3 and PRIMPOL had the same phenotypes. If the authors wish to suggest that these proteins function in the same pathway, they should test the epistasis of these proteins.

18. The data on siELL is confusing. Does depletion of ELL increase or decrease R-loops? One could make opposite predictions. Does it affect ROS-induced R-loops? Too many questions are unanswered!

19. MUS81 is known to promote fork restart in HU, so the data in Fig. 5d may not have anything to do with ROS.

20. The data in Fig. 5f actually argue the MUS81 and PRIMPOL promote fork progression independent of R-loops (when R-loops were removed by RNH1, MUS81 and PRIMPOL are still required for normal fork progression in HU). These results directly argue against the conclusion of this study.

21. The data in Fig. 5g is also confusing. RNH1 did not affect S1 cleavage. Does this mean that R-loops are not responsible for low HU induced ssDNA gaps? If so, ssDNA gaps are not relevant to the R-loop induced phenotypes induced by ROS?

22. The data in Fig. 6 are nice, but they are entirely expected. One has to assume that R-loops are a problem in HU for these experiments to have any novelty. Still, it is interesting to note how little NAC and RNH1 affected RPA/p-RPA signals. This means that although high HU also induces ROS and R-loops, they almost have no contributions to DNA breaks.

Reviewer #2 (Remarks to the Author):

Comments

This study by Andrs et al. provides evidence that reactive oxygen species (ROS)-induced replication fork slowing leads to replication fork reversal, presumably because of interference with transcription as a result of R-loop formation. The authors also demonstrated that prolonged replication arrest due to dNTPs depletion failed to promote fork reversal, but rather induced DNA breakage during S-phase in an R-loop-independent manner.

The study provides multiple lines of evidence to support the conclusions. This work will certainly establish a new understanding of the mechanisms whereby endogenous ROS-inducers such as oncogenes, ionizing radiation (IR), chemotherapeutic drugs, and metabolic agents promote genomic instability and tumor onset. While many of the experiments described support the evidence, additional experiments would strengthen this study as highlighted below:

1. The data presented in the Fig. 1 and the Extended data Fig. 1 demonstrate the role of ROS on global replication fork slowing. These effects were mimicked by TIMELESS depletion, which partially recapitulated its displacement from the replisome upon oxidative stress. Most of the experiments were conducted using 50uM H₂O₂ as an exogenous source of ROS. The authors should consider analyzing effects of other endogenous ROS inducers such as buthionine sulfoximine (BSO), oncogenic H-RasV12, or ionizing radiation (IR) followed by DNA fiber assays. One would expect the BSO treatment to mimic to some extent the conditions of endogenous oxidative stress.
2. The data presented in the figure 3 and the extended data fig.3 lack multiple panels. To illustrate the importance of ROS in interference with R-loop, authors should consider showing data related to NAC effects in each panel (3a-e). A similar argument applies to the experiments related to the genomic occupancy using RNH1 (D210N)-GFP IP and DRIP assay. The panels 3a, 3c, and 3f should include data on NAC effects to comprehensively illustrate the importance of ROS mitigation.
3. The data presented in Fig.5 provide a piece of evidence on the importance of MUS81 and PRIMPOL in the rescue of HU-induced fork slowing by RNase H1 overexpression. This experiment offers significant clues on whether MUS81 and PRIMPOL function as essential mediators. It would be highly informative to perform similar analyses in conditions of MUS81 and PRIMPOL overexpression.
4. In line with the results from prolonged exposure of cells to high concentrations of HU, the authors concluded that neither the addition of NAC, nor the overexpression of RNase H1 does show noticeable effects on DNA breakage induced by HU (gamma-H2AX and phosphorylation of RPA2). Given the relevance of NAC and RNase H1 to these phenomena, statistical analyses are needed to illustrate the data presented in figure 5 and extended data fig 6.
5. In the discussion, the authors arguably highlighted the importance of understanding the phenotypic consequences of ROS to replication stress, particularly in conditions of endogenous oxidative stress such as activation of RasV12, or metabolic stressors. This essential line of discussion aligns well with experiments requested above in point 1 (ie effects of BSO and H-RasV12). The findings will certainly demonstrate the physiological relevance of the use of H₂O₂ in the current study.

Reviewer #3 (Remarks to the Author):

In the manuscript entitled "Reactive oxygen species induce transcription-dependent replication stress in human cells", the authors show that increased levels of reactive oxygen species (ROS) induced by low concentrations of hydroxyurea (HU) promotes replication fork reversal. This leads to slowdown of fork progression that can be rescued by RNaseH1 overexpression or transcription inhibition, implying a role for active transcription and the formation of co-transcriptional R-loops as the mechanism to slow the fork. Additionally, high doses of HU (that lead to dNTP depletion) does not induce fork reversal and the persistent fork stalling results in DNA damage that is independent from the R-loop/transcription and therefore a distinct pathway.

In general, the data of the manuscript are of high quality and support the main conclusions of the manuscript and the results will be an important contribution to the field that should be reported in a

timely manner. However, a few major points should be addressed prior to publication:

1) Figure 1 is for the vast majority a recapitulation of the data from <https://doi.org/10.1038/s41467-020-19162-5> and DOI: 10.1126/science.aao3172 and thus do not represent novel findings. The authors may consider to move at least part of these data into a supplementary Figure.

2) The authors use NAC as a scavenger molecule to decrease ROS in the cells. Can the authors exclude the possibility that this small molecule has certain side effects in cells that could provide an alternative explanation for the observed effects? The authors may consider to either use a second independent small molecule to reproduce their results or an alternative could be to perform the experiments under conditions of hypoxia and therefore physically decrease the levels of ROS in cells without chemical intervention.

3) Figure 2/3: If ROS induce fork slowing via TIM dissociation and if HO-TRC mediated R-loops are a consequence of fork slowing, why does RNaseH1 overexpression rescue fork speed of siTIM to WT levels? (Fig2f-g) Additionally, RNaseH1 overexpression rescue fully fork speed and fork stalling in siTIM but only partially in HU/H₂O₂ treated cells suggesting that in the absence of TIM, the slowing down and the stalling of forks is fully dependent on R-loops but that in the context of ROS, R-loops only partially participate to those phenotype. It is understandable that ROS induce a pleiotropy of effects including R-loops and oxydated DNA bases which would explain the partial rescue but how do the authors explain the full R-loop dependency of fork slowing in siTIM?

4) ROS induce fork slowing and stalling, HO-TRC and R loops. siTIM have similar effects and have been implicated in the HU-ROS pathway in DOI: 10.1126/science.aao3172 and in Fig1. I think that the implication of fork speed and TIM in the results of Fig 2-3-4 would be stronger with a rescue of the phenotypes with cotreatment with COA instead of comparing siTIM side-by-side. Another alternative would be to show that siTIM and HU treatment do not show additive effects and are epistatic for that pathway.

5) Can the induction of fork slowing/stalling by ROS be an effect of TRCs produced by the sudden transcription of stress response genes or by oxidized bases? Can the R-loops/reversed forks be mapped to stress response genes? The authors could perform DRIP-qPCR analysis of genes that are expected to be upregulated upon oxidative stress.

6) In Fig 2k the authors use EM as an elegant approach to study and quantify fork reversal. Here, they use a treatment concentration of 500 μ M HU which influences both ROS levels and dNTP pools. Lower concentrations such as 50 μ M were similarly sufficient to produce ROS and cause fork slowdown/reversal phenotypes. Why did the authors choose 500 μ M, a concentration for which fork speed could not fully be rescued via NAC due to dNTP shortage issues (described like this for Fig. 1h)?

7) In Fig 3a-b the authors show an increase of PLA foci upon 50 μ M HU/H₂O₂ treatment, proposing that correspondingly induced ROS lead to TRCs and R-loops which would induce replication fork stalling. To test the specificity towards ROS as the causal event, can the authors perform the PLA experiments with NAC treatment? This way one could exclude that the effect of increased PLA foci occurs due to unspecific stressing of the cells upon HU/H₂O₂.

8) ChIP-qPCR analysis (Fig. 3f) of catalytically dead RNase H occupancy to map R-loop levels is performed upon 10 mM HU, a concentration vastly exceeding any previous treatments. Why is such a high concentration necessary when changes in TRC levels could be seen at 50 μ M and changes in R-loop levels at 4 mM?

9) It is still unclear to me if fork slowing is an effect of TRC or a cause for TRC. Fork slowing by siTIM induce R-loops in a slightly different fashion than HU/H₂O₂ (Fig3e and h). RNaseH1 overexpression rescue partially fork speed in HU/H₂O₂ treated cells and totally in TIM-depleted cells (Fig2a-g). This tends to place R-loops as a source of fork slowing down and not as the effect of fork slowing down. The authors should comment on a potential model in which ROS induce HO-TRCs which cause fork stalling and reversal and then the presence of stalled fork induce a global fork speed slow down.

10)

Minor:

1) Fig 3B: maybe add another color for TRC-negative cells (below threshold or something like that) to better see the difference from G1-S-G2 and see G2 persistent TRCs.

2) Fig 3 shows that HU increase of TRC in late S phase and slight increase of R loops in late S phase and in G2 and formation of R loops in early S phase. Those R-loops are located in early replicating genes. Fig 3 show also that siTIM show late S phase and G2 R-loops, a pattern that is not completely the same as HU. Does siTIM show formation of R loops at the same locus as HU? Or is it different?

3) The transition between the first two sections of Fig 1 (Fig 1 a-e and Fig 1f-i) feels quite poorly established and should be improved in order to facilitate the readability and understandability of the

manuscript. The first section focuses on fork reversal and nascent strand degradation can be driven by ROS in BRCA2-deficient cells. Next the authors show, that low HU/H₂O₂ doses cause the same effect. Subsequently, they switch to BRCA2-proficient cells which are used to demonstrate effects on fork velocity and symmetry. I feel like it should be more clear from the beginning what the purpose of studying BRCA2 proficient/deficient cells is. The authors should stress the role of BRCA2 in fork protection and why different BRCA2 backgrounds are needed to address fork degradation vs fork velocity effects.

4) In Fig S3a-b the authors show important controls for PLA specificity. It would be helpful to also add a quantification of PLA changes upon treatments in non-S-phase cells, further providing evidence for effect specificity to actively replicating cells. Particularly, the data of Fig. 3b also indicates slightly increased PLA foci levels in some G1 or G2/M cells upon HU/H₂O₂.

5) In Fig 3f it would help the readability to mark early vs late replicating genes in the figures or mention which ones belong to which category also in the figure description.

6) In the discussion the authors mention that in U2OS 20% of early S-phase origins reside within protein-coding genes and that the genome is pervasively transcribed (ll 241-243). This alone is not sufficient to conclude a frequent occurrence of HO TRCs. The residence of an origin within a gene does not necessitate a HO conflict. CD orientation is possible and probably the more likely case based on observations for a co-directional bias of transcription and replication.

7) The authors discuss new results in the discussion section, which should be avoided.

Point-by-Point Response to the Reviewers' comments (NCOMMS-22-27573-T)

We would like to thank the reviewers for taking the time to carefully assess our manuscript and for providing helpful and constructive feedback that helped us significantly improve the manuscript.

Answers to specific questions of reviewers:

Reviewer #1 (Remarks to the Author):

Previous studies suggested that low HU induces ROS to slowdown replication forks independently of dNTP depletion. In this study, the authors revisited the issue and suggested that ROS slows down replication forks by inducing co-transcriptional R-loops. They also present some data suggesting that several DNA repair/replication proteins are involved in the response to low HU. Finally, they show that R-loops are not responsible for the replication stress/DSBs induced by high HU. I really wanted to like this study, but I was quite disappointed as I read through the manuscript and figures. There are many issues that are not tested or convincingly addressed (see below). One major issue is that the model of this study is opposite from previous studies. Previous studies have suggested that replication fork reversal is a mechanism to slowdown forks. However, this study tries to argue that fork slowdown is the cause of fork reversal. This point is not well supported by the data. In addition, this study argues that replication slowdown increases R-loop-mediated TRCs, but direct evidence supporting this point is lacking. Although the model of this study is provocative, it is not convincing at this point. A much extended and reshaped study is needed for a paper in Nat Commun.

Response: We thank the reviewer for valuable comments that helped us improve our manuscript. We believe that the new data added to our manuscript provide further support for our model. Here, we would like to point out that we do not conclude that “ROS slow down replication forks by inducing co-transcriptional R-loops” or that “fork slowdown is caused by fork reversal” as stated above by the reviewer. Based on our data, we propose that ROS-induced fork slowing demonstrated by Jiri Lukas lab (DOI: 10.1126/science.aao3172) leads to R-loop-dependent fork reversal events (only a subset of forks that interfere with transcription). In other words, slowly moving forks due to TIMELESS-TIPIN complex dissociation induce R-loop formation after collision with an active transcription complex, which triggers fork reversal and thus completely stops DNA synthesis. Please note that replication fork slowing refers to active forks that move with a reduced velocity (e.g. forks lacking the fork acceleration factor TIMELESS-TIPIN), whereas fork reversal refers to inactive forks where the replisome is disrupted and the newly synthesized DNA strands anneal to each other to form the so-called “regressed arm”, which is covered by BRCA2-stabilized RAD51 filament. Thus, replication fork reversal is a mechanism that completely stops replication fork progression. This transaction is counteracted by RECQ1-mediated reverse branch migration leading to fork restart via the pathway (RECQ5-MUS81-LIG4-ELL-POLD3) we recently described (DOI: 10.1016/j.molcel.2019.10.026). In this manuscript, we demonstrate that PRIMPOL is also a component of this pathway.

Our model is also supported by the finding that R-loop-mediated fork stalling results upon partial inhibition of DNA replication by aphidicolin (APH), an inhibitor of replicative DNA polymerases, which is another way how to globally slow down the velocity of replication forks. We initially mentioned this finding only in the Discussion section. In the revised manuscript, it is described in a separate chapter of the Results section (page 9), and the results are shown in Fig. 5 and Suppl. Fig. 5. Moreover, we would like to emphasize here that we now present several lines of evidence suggesting that replication slowdown (fork progression with a reduced velocity) induces TRCs. Specifically, by PLA assay, we show that partial inhibition of DNA synthesis with low doses of APH increases colocalization between PCNA and the elongating form of RNAPII in U2OS cells (Fig. 5d,e). Furthermore, we show that colocalization between PCNA and RNAPII in cells treated with 50 μ M HU is suppressed by scavenging ROS with NAC or by PRDX2 depletion (Fig. 3c,i), which rescue ROS-induced replication slowdown (DOI: 10.1126/science.aao3172).

1. In the introduction, the authors should acknowledge that the slowdown of replication forks by co-transcriptional R-loops has been shown by James Manley's lab many years ago.

Response: The reviewer has probably in mind the following article: DOI: 10.1101/gad.17010011. We now cite this article in the first paragraph of the Discussion section along with the work from Philippe Pasero lab demonstrating that R-loop-mediated fork stalling is induced by TOP1 deficiency (DOI: 10.1038/ncb1984). Both these papers provide evidence that preexisting R-loops present a

roadblock to replication fork progression. In our work, we provide evidence that R-loops can also form as a consequence of a collision between an active transcription complex and a replisome progressing with a reduced velocity, having the same outcome as the preexisting R-loops: the blockage of fork progression by fork reversal.

2. The authors should also acknowledge that ROS-induced co-transcriptional R-loops have been reported and studied (Teng et al. Nat Commun 2018).

Response: Teng et al. (2018) found that locally produced ROS induced R-loop formation in a transcribed locus. This was accompanied by the generation of transcription-independent DNA damage revealed by accumulation of the DNA-damage marker γ H2AX at this locus. The authors proposed that ROS may induce R-loops by generating DNA single-strand breaks. This is consistent with the finding that a nick on the non-template DNA strand can serve as a strong R-loop initiation site in vitro (DOI:10.1128/MCB.00897-09). We now discuss the findings of Teng et al. (2018) in the Results section (page 7). However, several findings suggest that R-loops accumulating in HU-treated cells arise by a mechanism different from that proposed by Teng et al.: (i) Lukas lab demonstrated, and we could confirm, that exposure of cells to low HU, does not induce γ H2AX formation (Fig. S5c in DOI: 10.1126/science.aao3172); (ii) R-loops accumulate upon TIMELESS depletion (Fig. 3j in the revised manuscript); and most importantly, (iii) we now show that accumulation R-loops/RNH1(D210N)-GFP foci in HU-treated cells is suppressed by PRDX2 depletion, which should not interfere with the generation of ROS by HU (DOI: 10.1126/science.aao3172). These data are shown in Fig. 3h and Suppl. Fig. 3h and are described on page 7.

3. In Fig. 1e, it is not entirely clear whether nascent DNA degradation is the cause of sister fork asymmetry. MRE11 may cause fork collapse and contribute to this asymmetry.

Response: To our knowledge, replication fork collapse is rather caused by MRE11 deficiency (please see for example DOI: 10.1038/sj.emboj.7601045). It is well established that nascent DNA degradation induced by HU in BRCA2-depleted cells is mediated by MRE11 exonuclease in a manner dependent on fork reversal (DOI: 10.1038/s41467-017-01164-5). Sister fork asymmetry we observed upon prolonged exposure of BRCA2-depleted cells to HU (Fig. 1e) suggests that only one of the two sister forks has undergone nascent DNA resection: the fork which encounters an R-loop. This is supported by our finding that the sister fork asymmetry phenotype is rescued by the addition of mirin (Fig. 1e), an inhibitor of the exonuclease activity of MRE11 (DOI:10.1038/nchembio.63). It should be noted that mirin does not inhibit the endonuclease activity of MRE11 (DOI: 10.1016/j.molcel.2013.11.003), which would probably account for MRE11-mediated fork collapse in the model proposed by the reviewer. Moreover, 4 mM HU, used in our fork degradation experiments, blocks DNA synthesis (dNTP depletion), so the observed asymmetric pattern of sister replication tracts can only result from exonucleolytic activity. Please note that nascent DNA labelling (CldU/IdU) is performed before the addition of HU (Fig. 1a). Finally, we also found that the sister fork asymmetry phenotype of HU-treated cells could be rescued by ZRANB3 depletion, which blocks fork reversal, the prerequisite for MRE11-dependent nascent DNA degradation in BRCA2-deficient cells (please see Fig. R1).

Figure R1. Sister fork asymmetry generated after a 5-h exposure of BRCA2-depleted U2OS cells to 4 mM HU is rescued by ZRANB3 depletion. **(A)** Experimental workflow of DNA fiber assays. **(B)** Scatter plot of the values of IdU/CldU tract length ratio for indicated conditions. **(C)** Scatter plot of the values of the ratio between IdU tracts of sister replication forks (sister fork ratio; shorter tract/longer tract) for indicated conditions. Red lines indicate the median. **** $p < 0.0001$ (Mann-Whitney test).

4. The data in ED Fig. 1d and 1e are inconsistent. Exogenous nucleosides did not rescue sister fork asymmetry at all in 500 μ M HU, but partially rescued fork slowdown. Does this mean that fork slowdown

and sister fork asymmetry are at least partially independent phenotypes? The effects of nucleosides on “nascent DNA degradation” is not tested in high HU. Overall, the relationships among nascent DNA degradation, fork slowing, and sister fork asymmetry are not worked out convincingly.

Response: As discussed above, our data suggest that replication slowdown (decreased velocity of replication forks) is the cause of sister fork asymmetry (fork stalling). Exposure of cells to 500 μ M HU leads to both dNTP depletion and excessive ROS levels, the latter causing TIMELESS-TIPIN dissociation. Exogenous nucleosides partially restore normal fork speed by overcoming the lack of dNTPs, but do not rescue ROS-induced fork slowdown, which triggers R-loop formation and fork reversal upon head-on TRCs. Therefore, we still see sister fork asymmetry if exogenous nucleosides are added to cells treated with 500 μ M HU (the effect of 500 μ M HU + NUC corresponds to the effect of 50 μ M HU, which does not cause dNTP depletion).

The reviewer probably overlooked that we tested the effect of exogenous nucleosides on nascent DNA degradation in high HU (please see Fig. 1d,e in the original manuscript). We found that supplementation of BRCA2-depleted cells with exogenous nucleosides did not affect nascent DNA degradation induced by HU treatment, which is consistent with our model.

5. The authors heavily depend on NAC and Conoidin A for ROS-related conclusions, which is potentially problematic. These compounds could have unintended effects at the concentrations used. The use of low and high HU cannot “cleanly” separate ROS and dNTP depletion either. Can the authors confirm the results by depleting PRDX2 and RRM2 directly?

Response: As suggested by the reviewer, we tested the effect of PRDX2 depletion on replication slowdown and sister fork asymmetry in U2OS cells treated 50 μ M HU, as well as on HU-induced nascent DNA degradation in cells depleted of BRCA2. We obtained the same results as in the case of PRDX2 inhibition with Conoidin A. These new data are shown in Suppl. Fig. 1a-d,i-k in the revised manuscript. We also carried out experiments with another ROS scavenger (please see our response to point #2 of Reviewer 3).

We did not perform the suggested experiment with RRM2-depleted cells because it has been shown that RRM2 depletion leads to G1/S arrest (DOI:10.1371/journal.pone.0111714). Instead, we analyzed replication fork progression upon exposure of cells to L-buthionine-S,R-sulfoximine (BSO), an inhibitor of glutathione biosynthesis, causing ROS overproduction. This experiment was suggested by Reviewer 2 (point #1). We found that like HU or H₂O₂, BSO induced replication fork stalling (sister fork asymmetry) in a manner dependent on ROS and R-loop formation. These new data are shown in Suppl. Fig. 1f-h and Suppl. Fig. 2a-c.

6. The Timeless depletion experiments in Fig. 1J-n should be done in the presence and absence of low HU. If HU-induced ROS release Timeless from the replisome, Timeless knockdown should bypass the requirement for low HU to slowdown forks. It is not clear whether the effects of Timeless on fork speed and sister fork symmetry are specific to Timeless. Some other replisome proteins should be tested as controls.

Response: It was already shown by Jiri Lukas lab that low HU (50 μ M) did not further reduce replication fork velocity in TIMELESS-depleted cells (Fig. 1l in DOI: 10.1126/science.aao3172). We could confirm this finding. Moreover, we found that low HU did not further exacerbate replication fork stalling (sister fork asymmetry) and micronucleation caused by TIMELESS depletion. These data are shown in Suppl. Fig. 1t and Fig. 6f, respectively, in the revised manuscript.

The specificity of fork slowdown to TIMELESS-TIPIN depletion was also addressed by Lukas lab. They measured fork velocity upon knockdown of the components of the replication protection complex (RPC), which they found to copurify with nascent DNA in addition to the core replisome subunits. They observed a robust fork slow after depletion of TIMELESS, TIPIN and CLASPIN, respectively, but not AND-1 (DOI: 10.1126/science.aao3172; Fig. S3A). Since CLASPIN is an established mediator of the S-phase checkpoint, they reasoned that the reduced fork speed after CLASPIN depletion is related to a checkpoint defect and proposed that depletion of TIMELESS-TIPIN complex represents a “separation-of-function” condition to study the role of RPC proteins in replisome dynamics.

7. In Fig. 2a-d, the phenotypes of fork slowdown and sister fork asymmetry are inconsistent again. RNH1 only partially rescued fork slowdown, but fully rescued sister fork asymmetry. This raises further doubt on whether these phenotypes can be interpreted in the same way. It is also quite surprising that RNH1 fully rescued sister fork asymmetry in 2d, which means that dNTP depletion does not contribute to this phenotype at all. This should be retested by RRM2 depletion.

Response: As mentioned above, the reduction in fork velocity (reduction of DNA fiber length/IdU tracts in Fig. 2a-d) induced by HU is caused by three factors: (1) dNTP depletion (only in case of high HU);

(2) ROS-induced dissociation of the TIMELESS-TIPIN complex; and (3) R-loop-induced fork reversal, which is a consequence of the second factor. RNH1 over-expression restores normal fork progression only partially because it eliminates only the third factor (R-loops). In high HU (>500 μM), fork progression is still affected globally by dNTP depletion and TIMELESS-TIPIN dissociation, while in low HU (50 μM), only by TIMELESS-TIPIN dissociation. Sister fork asymmetry reflects R-loop-mediated fork stalling/reversal, therefore, a full rescue of this phenotype by RNH1 overexpression is seen not only in low but also in high HU (Fig. 2d). dNTP depletion does not contribute to the sister fork asymmetry phenotype. The effect of 50 μM H_2O_2 on DNA replication is like that of 50 μM HU (fork progression is compromised by ROS-induced dissociation of the TIMELESS-TIPIN complex and R-loop-induced fork reversal). We revised the text of the manuscript to make this point more clear. Please note that the scatter plots show the lengths of replication tracts of individual forks under given conditions. The red horizontal lines indicate the median. In HU, only a subset of forks is affected by TRC, while all forks are affected by dNTP depletion (only high HU) and ROS-induced TIMELESS-TIPIN dissociation.

8. The results in Fig. 2e-g are very surprising. The previous model suggests that the ROS-induced release of Timeless from the replisome is the “cause” of fork slowing. However, the data here suggest that R-loops, rather than the loss of Timeless, are the real cause of fork slowdown. If this is the case, why should PRDX2 matter? A replisome-associated ROS sensor should not affect ROS-induced co-transcriptional R-loops ahead of forks.

Response: The data in 2e-g show that R-loops cause replication fork stalling (detected as sister fork asymmetry) in TIMELESS-depleted cells. Please note that elimination of R-loops by RNH1 overexpression restores normal rate of fork progression only partially in TIMELESS-depleted cells; this is now more apparent from the new set of data presented in revised manuscript (Fig. 2f). In TIMELESS-depleted cells, fork progression is affected by two factors: (1) absence of TIMELESS, which globally reduces fork speed; and (2) R-loop-induced fork reversal (only forks that interfere with transcription). RNH1 overexpression eliminates only the second factor (R-loops). We added statistical analysis to Fig. 2f to demonstrate that upon RNH1 overexpression, the fork velocity (the median of replication tract lengths) in TIMELESS-proficient cells is higher than that in TIMELESS-depleted cells.

9. The experiments in Fig. 2h-j should be done in low HU. If low HU (and ROS) is sufficient to slowdown forks and cause fork reversal, it should also be sufficient to induce nascent DNA degradation in BRCA2 knockdown cells. If the effects of ROS can be reversed by RNH1, these experiments should work in low HU.

Response: Please note that in Suppl. Fig. 1a,b (now Suppl. Fig. 1e), we demonstrated that low HU (50 μM) or H_2O_2 (50 μM) induce nascent DNA degradation in BRCA2-depleted U2OS cells. In the revised manuscript, we added another set of data showing that these phenotypes are rescued by RNH1 overexpression (please see Suppl. Fig. 2p).

10. The EM experiments in Fig. 2k should also be done in low HU and in the absence of HU. If HU induces ROS to generate co-transcriptional R-loops, low HU should work identically as high HU. Furthermore, the inclusion of -HU controls is important for assessing how much RNH1/DRB/NAC suppress the HU effects.

Response: EM experiments are extremely time-consuming, and were performed on a collaborative basis in the Lopes lab. It is technically impossible and not realistic to significantly extend our EM datasets - as proposed by this reviewer - in the time frame of a standard revision. During experimental design, we carefully selected four conditions (high HU +/- RNH1/DRB/NAC) to address one of the central hypotheses of this manuscript, i.e. the contribution of oxidative stress and RNA:DNA hybrids in fork reversal. Reversed fork frequency in unperturbed U2OS cells has been reported multiple times by the Lopes lab to be around 5% (+/- 3%), thus in line with the frequency obtained with RNH1/DRB/NAC upon high HU treatment. We thus considered this control redundant and unnecessary to support key claims of this manuscript (as an example, the NT control was also not performed in a different publication from Cimprich and Lopes lab, i.e. Bai et al., Mol Cell 2020 (DOI: 10.1016/j.molcel.2020.04.031). Moreover, reversed fork frequency was reported to vary only marginally in 50 μM vs. 500 μM HU concentrations (please compare Fig. 1B in DOI: 10.1016/j.molcel.2020.04.031 with Fig. 3B in DOI: <https://doi.org/10.1083/jcb.201406099>). Therefore, as selecting one concentration was essential to produce reliable EM data in triplicate within a reasonable time frame, we selected the higher HU concentration to have both: (i) ROS production and (ii) dNTP depletion. We found that even upon higher HU concentration, fork reversal was largely dependent on ROS and R-loops, suggesting that replication arrest *per se* (dNTP depletion) does not play a major role in fork reversal. Given that HU-induced nascent DNA degradation in BRCA2-deficient cells is nowadays often used as a proxy assay

for fork reversal [see for example Berti et al., Nat Comms 2020 (DOI: 10.1038/s41467-020-17324-z); Liu et al., Sci Adv 2020 (DOI: 10.1126/sciadv.abc3598)], our finding that RNH1 overexpression rescues nascent DNA degradation induced in BRCRA2-defective cells by 50 μ M HU implies that fork reversal upon low HU also depends on R-loops (please see Suppl. Fig. 2p in the revised manuscript).

11. In Fig. 3c-e, both high and low HU should be tested. If low HU is sufficient to induce ROS and R-loops, it should work identically to high HU.

Response: We repeated these experiments also at an HU concentration of 50 μ M. As expected, we found that exposure of cells to 50 μ M HU induced the accumulation of RNH1(D210N)-GFP foci in cell nuclei in a manner dependent on ROS (rescue by NAC or PRDX2 depletion). We show these data in Suppl. Fig. 3c,d,i,j in the revised manuscript. Please note that in addition to the scatter plots showing the number of RNH1(D210N)-GFP foci in individual cells, we have now also included plots of the median values from 4 independent experiments for both high (4 mM; Fig. 3f and Suppl. Fig. 3h) and low (50 μ M; Suppl. Fig. 3d,j) HU to demonstrate the reproducibility of our findings.

12. Again, Fig. 3f should be done in low HU. If low HU is sufficient to induce R-loops, there is no reason to use 10 mM HU. Cells in early S phase may be particularly sensitive to the depletion of dNTPs.

Response: The ChIP-qPCR and DRIP-qPCR experiments presented in Fig. 3f and Suppl. Fig. 3d, respectively, are based on the work by Andre Nussenzweig lab showing that exposure of cells to 10 mM HU induces TRCs at early replicating genomic loci (defined as early replication fragile sites, ERFSS; DOI: 10.1016/j.cell.2013.01.006). We carried out these experiments long time ago to elucidate whether TRCs at these early replication loci lead to the accumulation of R-loops. We included these data in our manuscript to substantiate our finding of ROS-dependent accumulation of RNH1(D210N)-GFP foci in HU-treated cells, which indicate R-loop formation.

We have now repeated these experiments at an HU concentration of 4 mM and 50 μ M. In addition, NAC was added during these treatments to evaluate the role of ROS in the resulting accumulation of R-loops. We performed only DRIP-qPCR as this assay provides direct evidence for the presence of RNA:DNA hybrids in genomic DNA. We found that a 1-h treatment of cells with both concentrations of HU induced accumulation of RNA:DNA hybrids at the previously identified R-loop-prone regions of the APOE, BTBD19, RPL13A, BCL2 and GADD45A genes (DOI: 10.1101/gad.234070.113 and DOI: 10.1073/pnas.2114314119). Importantly, the observed accumulation of RNA:DNA hybrids at all these loci was suppressed by NAC, suggesting dependence on ROS. These data are now shown in Suppl. Fig. 3f,g and are described on page 6/7.

13. As mentioned above, it is surprising that loss of Timeless induces R-loops because Timeless travels with replication forks but not the transcription complex. Presumably depletion of Timeless should reduce fork speed and replication-transcription collisions, which may reduce R-loops. The authors should check whether loss of Timeless affects transcription. They should also check whether the R-loops in Timeless knockdown cells are co-transcriptional.

Response: The finding that TIMELESS depletion induces accumulation of R-loops is consistent with our model in which a collision of an impaired replication fork (fork progressing with a reduced velocity) with an active transcription complex leads to R-loop formation and subsequent fork reversal. This model is supported by numerous findings presented in this manuscript. Please note that we have also found that replication fork stalling (sister fork asymmetry) and micronucleation induced by TIMELESS depletion are rescued by overproduction of RNH1, providing further evidence that cells lacking TIMELESS accumulate R-loops (Fig. 2g and Fig. 6e).

As requested by the reviewer, we tested the effect of TIMELESS depletion on transcription in U2OS cells by measuring 5-fluorouridin (FU) incorporation into nascent RNA. QIBC analysis revealed that TIMELESS depletion reduced FU mean intensity in cell nuclei as compared to mock-depleted cells (Fig. R2A-C). This inhibitory effect is likely to be a consequence of elevated frequency of R-loop-mediated TRCs, which would attenuate transcription of the affected genes. In support of this notion, a similar reduction in FU-incorporation was seen if cells were treated with HU or H₂O₂ (Fig. R2C). Moreover, we observed a reduction in FU incorporation upon depletion of proteins that mediate the resolution of R-loop-mediated TRCs (data not shown). Exposure of cells to the transcription initiation inhibitor triptolide resulted in a much stronger inhibition of FU incorporation as compared to the effect of TIMELESS depletion or HU/H₂O₂ treatment (Fig. R2C).

To prove that TIMELESS knockdown induces accumulation of co-transcriptional R-loops, formation of RNH1(D210N)-GFP foci in TIMELESS-depleted cells was measured upon exposure to the transcription inhibitor triptolide, which was added to cells 3 h before pre-extraction and fixation. We found that triptolide substantially decreased the accumulation of RNH1(D210N)-GFP foci in TIMELESS-

depleted cells, suggesting that these cells accumulate co-transcriptional R-loops (Fig. R2D,E; please note that triptolide will not affect R-loops which are already formed). We are of the opinion that it is not necessary to show these data in the manuscript to support our conclusions.

Figure R2. (A-C) Effect of depletion of TIMELESS on transcription in U2OS cells. **(A)** Experimental workflow. RNA production was quantified using 5-fluorouridine (FU; 1 mM) pulse labeling. Cells were transfected for 24 h with 10 nM TIMELESS siRNA. Where required, cells were treated for 3 h with the transcription inhibitor triptolide (TRP; 1 μ M), for 30 min with 50 μ M H₂O₂ or for 60 min with 50 μ M HU. FU was added for the last 15 min. Cells were fixed and subjected to staining with anti-BrdU antibody (clone BU-33) followed by QIBC. **(B)** Western blot analysis of cell extracts. **(C)** Scatter plot of the values of FU intensity in cell nuclei obtained for indicated conditions (n \geq 1000). Red horizontal lines indicate the median. ****p < 0.0001 (Mann-Whitney test). **(D,E)** TIMELESS depletion induces formation of co-transcriptional R-loops. **(D)** Experimental workflow. TIMELESS was depleted in U2OS T-Rex [RNH1(D210N)-GFP] cells as in **(A)**. Expression of RNH1(D210N)-GFP was induced with Doxycycline (Dox) at a concentration of 1 ng/ml. **(E)** Scatter plot of the number of RNH1(D210N)-GFP foci in PCNA-positive cells upon depletion of TIMELESS and/or treatment with TRP. Red horizontal lines indicate the median. ns, not significant; ****p < 0.0001 (Mann-Whitney test).

14. Previous studies have shown that loss of fork reversal factors SMARCAL1/ZRANB3/HLTF increases fork speed in low HU or other conditions inducing fork reversal. If fork reversal is the consequence rather than the cause of fork slowdown as this study suggests, one would predict that loss of fork reversal should not alter fork slowdown, and the data from the previous studies would be difficult to explain. This is a key issue to be addressed by this study.

Response: Please note that in our experiments, abrogation of fork reversal (depletion of HLTF/ZRANB3 or inhibition of PARP1) restores normal replication velocity **only partially** in cells treated with 50 μ M HU (Fig. 1j-l, Fig. 4b, Suppl. Fig. 4b,c; replication tracts are still shorter compared to those in non-treated cells). We repeated these experiments several times and we always observed only a partial rescue. The same results (partial rescue of HU-induced fork slowing by HLTF depletion in U2OS cells) were recently reported by David Cortez lab (Fig. 4E in DOI: 10.1126/sciadv.abm0314). On the contrary, we found that the sister fork asymmetry in HU-treated cell (a measure of fork stalling/reversal) was completely rescued by abrogation of fork reversal by ZRANB3 depletion. These data, which are consistent with our model, are shown in Fig. 1m in the revised manuscript. **Thus, abrogation of fork reversal rescues fork stalling but not fork slowdown in HU-treated cells.** As we show in the manuscript, if fork reversal is prevented in cells treated with low HU, R-loop-stalled forks can be restarted via the MUS81-LIG4-PRIMPOL pathway, but such forks still move with a reduced velocity due to ROS-induced TIMELESS-TIPIN dissociation.

15. In Fig. 4, the effects of RNH1 on the cell cycle should be carefully monitored. A delay in cell cycle progression would affect all the phenotypes indirectly.

Response: As suggested, we performed cell cycle analysis of U2OS T-REX [RNH1(WT)-GFP] before and after induction of RNH1 expression. We found that RNH1 overexpression did not cause any significant delay in cell cycle progression (Fig. R3). Please note that the analyses of micronuclei and

53BP1 nuclear bodies in Fig. 4 (now Fig. 6) include only cells that have reached the G1 phase of the cell cycle. For micronucleus analysis, cells are blocked at cytokinesis by cytochalasin B (only binucleated cells are scored for the presence of micronuclei); for 53BP1 NB analysis, G1 cells are identified through absence of Cyclin A.

16. The data in Fig. 5 support the notion that fork reversal causes fork slowdown, but not the idea that fork reversal is a consequence of fork slowdown. The author switched the concepts when they moved from Fig. 4 to Fig. 5. This is not helpful to the overall conclusion of this study.

Response: We did not switch the concept when moving from Fig. 4 to Fig. 5. As mentioned above (point #14), the data presented in Fig. 5 (now Fig. 4) show that abrogation of fork reversal restores normal fork progression **only partially** (the average length of the IdU tracts upon these conditions is still significantly smaller than that in non-treated cells; replication slowdown persists). In contrast, depletion of ZRANB3 fully prevents HU-induced sister fork asymmetry (Fig. 1m), supporting our conclusion that upon replication slowdown, fork progression is additionally hampered by fork reversal (only forks which collide with transcription).

17. It is surprising that MUS81/LIG4/POLD3 and PRIMPOL had the same phenotypes. If the authors wish to suggest that these proteins function in the same pathway, they should test the epistasis of these proteins.

Response: We performed the suggested epistasis analysis using MUS81 knockout HeLa Kyoto cells and the wild-type counterparts, which were also used in our previous studies (DOI: 10.1016/j.molcel.2019.10.026). We found that PRIMPOL knockdown did not further increase the percentage of stalled forks in MUS81 knockout cells after release from HU arrest, although it induced persistent fork stalling in the parental HeLa Kyoto cells to a level similar to that caused by MUS81 knockout alone. These results, which are now shown in Fig. 4e-g and described on page 8, suggest that MUS81 and PRIMPOL act in the same replication restart pathway.

18. The data on siELL is confusing. Does depletion of ELL increase or decrease R-loops? One could make opposite predictions. Does it affect ROS-induced R-loops? Too many questions are unanswered!

Response: This manuscript does not aim to dissect the role of the transcription elongation factor ELL in restarting R-loop-stalled forks. In fact, this was the subject of our recently published study, which suggested that ELL may promote transcription restart at sites of R-loop-mediated TRCs to allow for replication restart (DOI: 10.1016/j.molcel.2019.10.026). Importantly, our epistasis analysis demonstrated that ELL and MUS81 act in a common pathway to counteract R-loop-mediated replication stress (please see Fig. 6G in DOI: 10.1016/j.molcel.2019.10.026).

19. MUS81 is known to promote fork restart in HU, so the data in Fig. 5d may not have anything to do with ROS.

Response: To address this point, we investigated whether MUS81 is required for fork restart if HU is combined with the ROS scavenger N-acetyl cysteine (NAC). We found that upon these conditions, MUS81 depletion did not impair replication restart after release from HU arrest. These data, which are now shown in Fig. 4h and described on page 8, support our conclusion that MUS81 mediates replication restart at ROS-induced R-loops in cell exposed to HU. If HU-induced ROS are eliminated (NAC), fork can efficiently restart without MUS81 after removal of HU (there is no roadblock to fork progression, fork restart depends just on dNTP regeneration).

20. The data in Fig. 5f actually argue the MUS81 and PRIMPOL promote fork progression independent of R-loops (when R-loops were removed by RNH1, MUS81 and PRIMPOL are still required for normal fork progression in HU). These results directly argue against the conclusion of this study.

Response: The results presented in Fig. 5f (now Fig. 4j) support our model wherein R-loop is formed after a collision of an active transcription complexes with a slowly moving replication fork, and this in turn leads to fork reversal. If R-loops are eliminated by overproduction of RNH1, fork reversal does not take place, but the restoration of DNA synthesis requires MUS81 and PRIMPOL because fork progression is still impaired by the transcription complex, which may remain on the DNA after R-loop removal. We discuss this possible scenario in the revised manuscript (page 8/9). In contrast, RECQ1 was not required for fork restart upon these conditions (overexpression of RNH1), which implies that fork reversal did not take place. Thus, the full sequence of events is as follows: TRC → R-loop formation → fork reversal ↔ RECQ1-mediated reverse branch migration → R-loop removal/transcription restart (ELL) → replication restart via MUS81-LIG4-PRIMPOL pathway (please see also our Mol Cell paper; DOI: 10.1016/j.molcel.2019.10.026).

To verify our conclusions, we investigated whether MUS81 and PRIMPOL were required for fork progression if HU was combined with the transcription initiation inhibitor triptolide. In our manuscript, we demonstrate that like RNH1 overexpression, also transcription inhibition (initiation or elongation) partially rescues normal fork progression in cells treated with 50 μM HU (now Suppl. Fig. 2d-I). Importantly, we found that in this case (transcription inhibitor+HU), the rescue of fork progression was independent of MUS81 and PRIMPOL as predicted by our model (inhibition of transcription prevents TRCs). These data are now shown in Suppl. Fig. 4e and discussed on page 8 in the revised manuscript.

21. The data in Fig. 5g is also confusing. RNH1 did not affect S1 cleavage. Does this mean that R-loops are not responsible for low HU induced ssDNA gaps? If so, ssDNA gaps are not relevant to the R-loop induced phenotypes induced by ROS?

Response: The data in Fig. 5g (now Fig. 4k) show that RNH1 overexpression **promotes** S1 nuclease cleavage (shortening of IdU tracts of HU-treated cells by S1 endonuclease). The same effect is seen upon ZRANB3 depletion (prevention of fork reversal) as previously reported (DOI: 10.1016/j.molcel.2020.04.031). In cells exposed to 50 μM HU, ssDNA gaps are generated during PRIMPOL-mediated DNA synthesis, which is induced by R-loop removal (RNH1 overexpression) or by inhibition of fork reversal (ZRANB3 depletion) as discussed above in the point # 20. Thus, elimination of R-loops or inhibition of R-loop-induced fork reversal in HU-treated cells triggers MUS81-PRIMPOL-mediated DNA synthesis associated with ssDNA gaps. Again, this is prevented by inhibition of transcription (no S1 nuclease cleavage is seen if HU is combined with transcription inhibitor).

22. The data in Fig. 6 are nice, but they are entirely expected. One has to assume that R-loops are a problem in HU for these experiments to have any novelty. Still, it is interesting to note how little NAC and RNH1 affected RPA/p-RPA signals. This means that although high HU also induces ROS and R-loops, they almost have no contributions to DNA breaks.

Response: This is the point that we want to make: ROS and R-loops do not contribute to DNA breakage in S-phase upon prolonged arrest of DNA synthesis by high HU (dNTP depletion/RPA exhaustion). However, as we discuss in the Discussion section, persistent R-loop-stalled forks in cells entering mitosis (e.g. in low HU or upon TIMELESS depletion) lead to DNA breakage during chromosome segregation due to the presence of under-replicated regions. This is manifested by micronucleation and formation of 53BP1 nuclear bodies in nascent G1 cells. Of note, such DNA breaks in G1 can lead to chromosomal rearrangements found in cancers (DOI: 10.1016/j.cell.2013.01.006).

R-loop-stalled forks are protected in S-phase by fork reversal and assembly of BRCA2-stabilised RAD51 filament on the regressed arm, which prevents generation of ssDNA regions and fork collapse by nucleases in S/G2. Therefore, ROS-induced R-loops in HU-treated cells do not contribute to the observed S-phase-specific DNA breakage in high HU. As discussed, we need a conflict with an active transcription complex for fork reversal to occur. Arrest of fork progression by dNTP depletion does not trigger fork reversal, and if this happens before a conflict with transcription in non-transcribed regions, it leads to accumulation of unprotected forks (no fork reversal, ssDNA regions without RPA), which are prone to breakage by nucleases in S-phase.

Reviewer #2 (Remarks to the Author):

This study by Andrs et al. provides evidence that reactive oxygen species (ROS)-induced replication fork slowing leads to replication fork reversal, presumably because of interference with transcription as a result of R-loop formation. The authors also demonstrated that prolonged replication arrest due to dNTPs depletion failed to promote fork reversal, but rather induced DNA breakage during S-phase in an R-loop-independent manner.

The study provides multiple lines of evidence to support the conclusions. This work will certainly establish a new understanding of the mechanisms whereby endogenous ROS-inducers such as oncogenes, ionizing radiation (IR), chemotherapeutic drugs, and metabolic agents promote genomic instability and tumor onset. While many of the experiments described support the evidence, additional experiments would strengthen this study as highlighted below:

Response: We are glad that this reviewer finds our manuscript to be of interest to the field. We would like to thank the reviewer for valuable comments that helped us improve the manuscript.

1. The data presented in the Fig. 1 and the Extended data Fig. 1 demonstrate the role of ROS on global replication fork slowing. These effects were mimicked by TIMELESS depletion, which partially recapitulated its displacement from the replisome upon oxidative stress. Most of the experiments were conducted using 50 μ M H₂O₂ as an exogenous source of ROS. The authors should consider analyzing effects of other endogenous ROS inducers such as buthionine sulfoximine (BSO), oncogenic H-Rasv12, or ionizing radiation (IR) followed by DNA fiber assays. One would expect the BSO treatment to mimic to some extent the conditions of endogenous oxidative stress.

Response: As suggested by the reviewer, we analyzed the effect of BSO on the progression of replication forks in U2OS cells. We found that similarly to HU or H₂O₂, BSO induced ROS- and R-loop-dependent fork stalling (sister fork asymmetry). These data are shown in Suppl. Fig. 1f-h and Suppl. Fig. 2a-c in the revised manuscript. Of note, as in case of low HU or H₂O₂, elimination of R-loops by RNH1 overexpression restored fork progression only partially in BSO-treated cells (Suppl. Fig. 2a,b), whereas a nearly full rescue of fork progression was observed upon scavenging ROS with NAC (Suppl. Fig. 1 f,g). This supports our proposal that two factors affect fork progression in cells with excessive levels of ROS: (i) TIMLESS-TIPIN dissociation from the replisome (a global reduction in fork velocity); and (ii) R-loop-induced fork stalling (only forks interfering with transcription), which is a consequence of the global replication slowdown.

We also study the molecular basis of HRASV12-induced replication stress (an ongoing project). We found that HRASV12 overexpression induces R-loop accumulation and R-loop-dependent fork stalling and micronucleation in U2OS cells. Importantly, our preliminary data suggest that these phenotypes also depend on ROS. HRASV12 is known to increase the cellular levels of ROS by inducing the expression of NADPH oxidase 4 (DOI: 10.1038/onc.2011.327).

2. The data presented in the figure 3 and the extended data fig.3 lack multiple panels. To illustrate the importance of ROS in interference with R-loop, authors should consider showing data related to NAC effects in each panel (3a-e). A similar argument applies to the experiments related to the genomic occupancy using RNH1 (D210N)-GFP IP and DRIP assay. The panels 3a, 3c, and 3f should include data on NAC effects to comprehensively illustrate the importance of ROS mitigation.

Response: As suggested, we included NAC treatment also in the PLA and DRIP-qPCR experiments. We found that NAC largely suppressed HU-induced TRCs (RNAPII/PCNA colocalization; PLA assay) and accumulation of RNA:DNA hybrids at R-loop-prone loci (DRIP-qPCR). These data are now shown in Fig. 3c and Suppl. Fig. 3f,g, respectively, and described on page 6/7. For the sake of clarity, we do not show the RNH1(D210N)-GFP ChIP-qPCR data in the revised manuscript.

Quantification of PLA and RNH1(D210N)-GFP foci was done automatically using Olympus ScanR software. As seen from the scatter plots, there is a broad distribution of the number of foci in cells for each condition, therefore, it is difficult and counterproductive to show any representative images. The images used in the manuscript are just to illustrate the "positive" and "negative" cells.

3. The data presented in Fig.5 provide a piece of evidence on the importance of MUS81 and PRIMPOL in the rescue of HU-induced fork slowing by RNase H1 overexpression. This experiment offers significant clues on whether MUS81 and PRIMPOL function as essential mediators. It would be highly informative to perform similar analyses in conditions of MUS81 and PRIMPOL overexpression.

Response: We thank the reviewer for this interesting idea. In our Mol Cell 2020 paper (DOI: 10.1016/j.molcel.2019.10.026), we provide evidence suggesting that RECQ5 DNA helicase mediates the switch between fork reversal and fork restart at R-loops by removing RAD51 recombinase from the

fork junction to prevent fork reversal and to facilitate fork cleavage by MUS81-EME1 endonuclease, which triggers replication restart. Therefore, we thought to test the effect of overexpression of this component of the MUS81-LIG4-PRIMPOL axis on replication perturbation induced low HU (50 μ M). For this, we used a U2OS T-Rex cell line where overexpression of a RECQ5 transgene can be induced by doxycycline (DOI: 10.1016/j.molcel.2019.10.026). We found that RECQ5 overexpression could partially restore normal fork progression and completely prevented sister fork asymmetry (fork stalling) in cells exposed to low HU (please see Fig. R4 below). These findings are consistent with our model of RECQ5 acting as a molecular switch between R-loop-induced fork stalling and replication restart. Please note that RECQ5 overexpression restored normal fork progression only partially because it counteracts R-loop-induced fork reversal but not ROS-induced TIMELESS-TIPIN dissociation from the replisome, which globally reduces fork velocity. As there is no doubt that these data provide insight into the molecular mechanism involved in restarting R-loop-stalled forks, we are of the opinion that including them in the current manuscript would not provide any further support for our conclusions. The point of the data in Fig. 5 (now Fig. 4) is to provide additional evidence for our proposal that HU induces R-loop-dependent replication stress (reactivation of HU-stalled fork is dependent upon proteins which are implicated in replication restart at R-loops). We also want to emphasize that PRIMPOL is a component of this pathway.

Figure R4. RECQ5 overexpression restores replication fork progression in HU-treated cells. **(A)** Experimental workflow of DNA fiber assay. U2OS T-Rex cells harbouring a RECQ5 transgene under control of doxycyclin-regulated CMV promoter were cultured in the absence or the presence of 0.5 ng/ml doxycyclin for 48 h, followed by labeling the replication tracts with CldU and IdU. HU was present in a concentration of 50 μ M during IdU labeling. **(B)** Western blot analysis of cell extracts. MSH6 was selected as a loading control. **(C)** Scatter plot of the values of IdU/CldU tract length ratio for indicated conditions. Red lines indicate the median ($n > 400$). **** $p < 0.0001$ (Mann-Whitney test). **(D)** Scatter plot of the values of sister fork ratio for indicated conditions. Red lines indicate the median ($n > 100$). ns, not significant; **** $p < 0.0001$ (Mann-Whitney test).

4. In line with the results from prolonged exposure of cells to high concentrations of HU, the authors concluded that neither the addition of NAC, nor the overexpression of RNase H1 does show noticeable effects on DNA breakage induced by HU (gamma-H2AX and phosphorylation of RPA2). Given the relevance of NAC and RNase H1 to these phenomena, statistical analyses are needed to illustrate the data presented in figure 5 and extended data fig 6.

Response: Statistical analysis data were added to these figures (now Fig. 7 and Suppl. Fig. 7).

5. In the discussion, the authors arguably highlighted the importance of understanding the phenotypic consequences of ROS to replication stress, particularly in conditions of endogenous oxidative stress such as activation of RasV12, or metabolic stressors. This essential line of discussion aligns well with experiments requested above in point 1 (ie effects of BSO and H-RasV12). The findings will certainly demonstrate the physiological relevance of the use of H₂O₂ in the current study.

Response: As mentioned above (response to point #1 of Reviewer 2), the data showing R-loop-dependent fork stalling in cells treated with BSO were included in the manuscript to demonstrate the physiological relevance of our findings.

Reviewer #3 (Remarks to the Author):

In the manuscript entitled “Reactive oxygen species induce transcription-dependent replication stress in human cells”, the authors show that increased levels of reactive oxygen species (ROS) induced by low concentrations of hydroxyurea (HU) promotes replication fork reversal. This leads to slowdown of fork progression that can be rescued by RNAseH1 overexpression or transcription inhibition, implying a role for active transcription and the formation of co-transcriptional R-loops as the mechanism to slow the fork. Additionally, high doses of HU (that lead to dNTP depletion) does not induce fork reversal and the persistent fork stalling results in DNA damage that is independent from the R-loop/transcription and therefore a distinct pathway.

In general, the data of the manuscript are of high quality and support the main conclusions of the manuscript and the results will be an important contribution to the field that should be reported in a timely manner. However, a few major points should be addressed prior to publication:

Response: We are glad that this reviewer finds our results to be of high quality and importance to the field. We would like to thank the reviewer for valuable suggestions on how to improve our manuscript.

1) Figure 1 is for the vast majority a recapitulation of the data from <https://doi.org/10.1038/s41467-020-19162-5> and DOI: 10.1126/science.aao3172 and thus do not represent novel findings. The authors may consider to move at least part of these data into a supplementary Figure.

Response: It is true that Jiri Lukas lab already reported (DOI:10.17632/h9rmg6hc9k.1) that HU-induced nascent DNA degradation in BRCA2-depleted cells depends on ROS, but the authors concluded that ROS trigger this phenomenon by promoting phosphorylation of MRE11. In Fig. 1, we demonstrate that nascent DNA degradation in BRCA2-depleted cells is fully suppressed not only by the ROS scavenger NAC but also by inhibiting PRDX2, suggesting that the underlying molecular mechanism also involves ROS-induced replication slowdown. Moreover, earlier work by Jiri Lukas lab (DOI: 10.1126/science.aao3172) did not demonstrate that HU and H₂O₂ induce sister fork asymmetry, which indicates fork stalling. For this reason, we prefer to keep the panels a-i in the main figure. We moved to Suppl. Fig. 1 the panels j-n showing sister fork asymmetry in TIMELESS depleted cells, which was previously reported (<https://doi.org/10.1038/s41467-020-19162-5>).

2) The authors use NAC as a scavenger molecule to decrease ROS in the cells. Can the authors exclude the possibility that this small molecule has certain side effects in cells that could provide an alternative explanation for the observed effects? The authors may consider to either use a second independent small molecule to reproduce their results or an alternative could be to perform the experiments under conditions of hypoxia and therefore physically decrease the levels of ROS in cells without chemical intervention.

Response: We think that it is unlikely that the results we obtained with NAC (e.g. rescue of HU-induced fork slowing/stalling) are due to some side effects of this small molecule. Please note that essentially the same results were obtained with the PRDX2 (ROS sensor) inhibitor Conoidin A, or upon PRDX2 depletion (please see for example Fig. 1d,e,h,i and Suppl. Fig. 1a-d,i-k). Nevertheless, we repeated some of our DNA fiber experiments with another ROS scavenger:

Mn(III)tetrakis(4-benzoic acid) porphyrin chloride (MnTBAP), a cell-permeable synthetic antioxidant with superoxide dismutase-like activity (Sigma-Aldrich). As shown in Fig. R5, we found that MnTBAP could rescue sister fork asymmetry in U2OS cells treated with 50 μM HU or 50 μM H₂O₂.

Figure R5. Elimination of ROS prevents replication fork stalling in U2OS cells exposed to hydroxyurea or hydrogen peroxide. Top: Experimental workflow of DNA fiber assays. The ROS scavenger Mn(III)tetrakis(4-benzoic acid) porphyrin chloride (MnTBAP) was added 1 h before replication tract labeling with CldU and IdU. Hydroxyurea (HU) or hydrogen peroxide (H₂O₂) were present in a concentration of 50 μM during IdU labeling. Bottom: Scatter plot of the values of sister fork ratio for indicated conditions. Red lines indicate the median (n > 100). ns, not significant; ****p < 0.0001 (Mann-Whitney test). Asymmetric pattern of sister replication tracts indicates fork stalling.

As suggested by the reviewer #2, we analyzed the effect of another ROS producer, buthionine sulfoximine (BSO), on the progression of replication forks in U2OS cells. We found that similarly to HU or H₂O₂, BSO induced ROS- and R-loop-dependent fork stalling (sister fork asymmetry). These data are shown in Suppl. Fig. 1f-h and Suppl. Fig. 2a-c in the revised manuscript. Surprisingly, hypoxia rather increases ROS levels in the cell (DOI:10.17632/h9rmg6hc9k.1). Moreover, hypoxia has been shown to induce replication fork stalling in absence of detectable DNA damage (DOI: 10.1016/j.molcel.2013.10.019). An ongoing study in our lab addresses whether hypoxia induces R-loop-mediated fork stalling/reversal.

3) Figure 2/3: If ROS induce fork slowing via TIM dissociation and if HO-TRC mediated R-loops are a consequence of fork slowing, why does RNaseH1 overexpression rescue fork speed of siTIM to WT levels? (Fig2f-g) Additionally, RNaseH1 overexpression rescue fully fork speed and fork stalling in siTIM but only partially in HU/H₂O₂ treated cells suggesting that in the absence of TIM, the slowing down and the stalling of forks is fully dependent on R-loops but that in the context of ROS, R-loops only partially participate to those phenotype. It is understandable that ROS induce a pleiotropy of effects including R-loops and oxydated DNA bases which would explain the partial rescue but how do the authors explain the full R-loop dependency of fork slowing in siTIM?

Response: Please note that RNH1 overexpression rescues fork speed only partially in TIMELESS-depleted cells, although in case of HU/H₂O₂-treated cells (Fig. 2c), the partial rescue is more apparent. Only a partial rescue of fork speed was also seen if TIMELESS-depleted cells were treated with the transcription elongation inhibitor DRB (please see Suppl. Fig. 2n in the revised manuscript). To confirm the data shown in Fig. 2f, we repeated the experiment. The results are show in revised manuscript and include statistical analysis (Fig. 2f). Again, we observed only a partial rescue of fork speed upon RNH1 overexpression in TIMELESS-depleted cells.

4) ROS induce fork slowing and stalling, HO-TRC and R loops. siTIM have similar effects and have been implicated in the HU-ROS pathway in DOI: 10.1126/science.aao3172 and in Fig1. I think that the implication of fork speed and TIM in the results of Fig 2-3-4 would be stronger with a rescue of the phenotypes with cotreatment with COA instead of comparing siTIM side-by-side. Another alternative would be to show that siTIM and HU treatment do not show additive effects and are epistatic for that pathway.

Response: As mentioned in our response to Reviewer 1 (point #6), it was already shown by Jiri Lukas lab that low HU (50 μ M) did not further reduce replication fork velocity in TIMELESS-depleted cells (Fig. 1l in DOI: 10.1126/science.aao3172). We could confirm this finding. Moreover, we found that low HU did not further exacerbate replication fork stalling (sister fork asymmetry) and micronucleation caused by TIMELESS depletion. These data are shown in Suppl. Fig. 1t and Fig. 6f, respectively, in the revised manuscript.

5) Can the induction of fork slowing/stalling by ROS be an effect of TRCs produced by the sudden transcription of stress response genes or by oxidized bases? Can the R-loops/reversed forks be mapped to stress response genes? The authors could perform DRIP-qPCR analysis of genes that are expected to be upregulated upon oxidative stress.

Response: We think that this scenario is unlikely because we see R-loop-dependent fork stalling also in TIMELESS-depleted cells or upon APH-induced replication slowdown. Moreover, HU- as well as H₂O₂-induced fork stalling was suppressed by PRDX2 inhibition/depletion, which does reduce the excessive ROS levels in these cells. Nevertheless, using U2OS cells, we tested the effect of HU and H₂O₂ on transcription of the GPX1 and SETX genes, which are known to be upregulated upon oxidative stress (DOI: 10.1016/B978-0-12-405881-1.00013-6 for GPX1; DOI: 10.1038/s41467-021-24066-z for SETX). We found that a 1-h treatment with these ROS-generating agents (conditions used in our DNA fiber experiments) did not increase transcription of the GPX1 and SETX genes in U2OS cells. Elevated expression of these genes was detected only after prolonged exposure (24 h) of cells to 4 mM HU (Fig. R6). We also would like to point out that HU-treatment does not lead to an immediate generation of DNA damage (gamma-H2AX) in U2OS cells (please see Fig. S5c in DOI: 10.1126/science.aao3172).

Figure R6. Effect of HU and H₂O₂ on the expression of GPX1 and SETX genes. U2OS cells were exposed to the indicated concentrations of HU or H₂O₂ for 1 h or 24 h. Following these treatments, total RNA was extracted and cDNA was synthesised to quantify specific mRNA levels by RT-qPCR. For each gene (GPX1, SETX), relative mRNA levels were determined using GAPDH as the reference gene. The fold change in expression upon treatments as compared to the non-treated sample is shown. Data are mean ± SEM (n=3).

6) In Fig 2k the authors use EM as an elegant approach to study and quantify fork reversal. Here, they use a treatment concentration of 500 µM HU which influences both ROS levels and dNTP pools. Lower concentrations such as 50 µM were similarly sufficient to produce ROS and cause fork slowdown/reversal phenotypes. Why did the authors choose 500 µM, a concentration for which fork speed could not fully be rescued via NAC due to dNTP shortage issues (described like this for Fig. 1h)?

Response: Please see our response to the point # 10 of Reviewer 1.

7) In Fig 3a-b the authors show an increase of PLA foci upon 50 µM HU/H₂O₂ treatment, proposing that correspondingly induced ROS lead to TRCs and R-loops which would induce replication fork stalling. To test the specificity towards ROS as the causal event, can the authors perform the PLA experiments with NAC treatment? This way one could exclude that the effect of increased PLA foci occurs due to unspecific stressing of the cells upon HU/H₂O₂.

Response: We performed the suggested PLA experiment with NAC treatment. We found that addition of this ROS scavenger largely prevented the formation of PLA foci between PCNA and RNAPII in U2OS cells treated with 50 µM HU (please see Fig. 3c in the revised manuscript). Moreover, we now also show that the formation of these PLA foci in HU-treated cells was abrogated by depletion of PRDX2 (Fig. 3i), suggesting that persistent TRCs result as a consequence of replisome slowdown induced by ROS.

8) ChIP-qPCR analysis (Fig. 3f) of catalytically dead RNase H occupancy to map R-loop levels is performed upon 10 mM HU, a concentration vastly exceeding any previous treatments. Why is such a high concentration necessary when changes in TRC levels could be seen at 50 µM and changes in R-loop levels at 4 mM?

Response: Please see our response to point #12 of Reviewer 1.

9) It is still unclear to me if fork slowing is an effect of TRC or a cause for TRC. Fork slowing by siTIM induce R-loops in a slightly different fashion than HU/H₂O₂ (Fig3e and h). RNaseH1 overexpression rescue partially fork speed in HU/H₂O₂ treated cells and totally in TIM-depleted cells (Fig2a-g). This tends to place R-loops as a source of fork slowing down and not as the effect of fork slowing down. The authors should comment on a potential model in which ROS induce HO-TRCs which cause fork stalling and reversal and then the presence of stalled fork induce a global fork speed slow down.

Response: We again would like to point out that the rescue of fork progression in TIMELESS-depleted cells by RNH1 overexpression or transcription inhibition is only partial (please see our response to the point #3).

The model suggested by the reviewer (ROS-induced TRCs lead to a global fork speed slowdown) is not consistent with our finding that the impaired fork progression upon exposure of cells to low HU or H₂O₂ is clearly only partially rescued by RNH1 overexpression or transcription inhibition. Rescue is also only partial upon abrogation of fork reversal (depletion of ZRANB3/HLTF), which depends on

transcription and R-loop formation. Therefore, we think that rather ROS-induced fork slowing - an active fork moving with a reduced velocity due to TIMELESS-TIPIN dissociation - causes R-loop formation during a transcription-replication collision, which induces fork reversal and hence stops DNA synthesis at the affected forks. This model is also supported by the finding that R-loop-dependent fork stalling is induced by partial inhibition of replicative DNA polymerases with low doses of APH, which also globally reduces replication fork velocity. These data are now shown in Fig. 5 and Suppl. Fig. 5.

Minor:

1) Fig 3B: maybe add another color for TRC-negative cells (below threshold or something like that) to better see the difference from G1-S-G2 and see G2 persistent TRCs.

Response: It is difficult to set a threshold to separate TRC-positive and -negative cells in the scatter plots in Fig. 3b. There are also PLA foci in G1/G2 cells, but we can clearly see that the increase in cells with a high number of foci after HU/H₂O₂ treatment is highly specific to S-phase cells (as seen in the bottom scatter plot).

2) Fig 3 shows that HU increase of TRC in late S phase and slight increase of R loops in late S phase and in G2 and formation of R loops in early S phase. Those R-loops are located in early replicating genes. Fig 3 show also that siTIM show late S phase and G2 R-loops, a pattern that is not completely the same as HU. Does siTIM show formation of R loops at the same locus as HU? Or is it different?

Response: We are of the opinion the QIBC analyses of R-loop levels [number of RNH1(D210N)-GFP foci] in the populations HU-treated or TIMELESS-depleted cells, shown in Fig. 3e (now Fig. 3g) and Fig. 3h (now Fig. 3k), respectively, do not reveal any significant differences with respect to the cell cycle stages where R-loops are preferentially formed. Please note that TIMELESS is continuously depleted over 24 hours but HU is present only for 1 hour. This factor will contribute to different distribution of R-loops over the cell cycle. To conclusively determine whether TIMELESS depletion induces R-loop formation in the same loci as HU, we would need to perform a series of DRIP-sequencing experiments. We believe that the reviewer agrees that the resolution of this question is not essential to support our conclusions. Some indirect evidence suggesting that TIMELESS depletion and HU treatment induce R-loop formation at the same regions is provided by our new finding that low HU did not exacerbate replication fork stalling and micronucleation in TIMELESS-depleted cells, which depends on R-loops (please see Suppl. Fig. 1t and Fig. 6f).

3) The transition between the first two sections of Fig 1 (Fig 1 a-e and Fig 1f-i) feels quite poorly established and should be improved in order to facilitate the readability and understandability of the manuscript. The first section focuses on fork reversal and nascent strand degradation can be driven by ROS in BRCA2-deficient cells. Next the authors show, that low HU/H₂O₂ doses cause the same effect. Subsequently, they switch to BRCA2-proficient cells which are used to demonstrate effects on fork velocity and symmetry. I feel like it should be more clear from the beginning what the purpose of studying BRCA2 proficient/deficient cells is. The authors should stress the role of BRCA2 in fork protection and why different BRCA2 backgrounds are needed to address fork degradation vs fork velocity effects.

Response: We tried to improve this part of the manuscript. The key point of Fig. 1 and Suppl. Fig. 1 is to demonstrate that ROS-induced replication slowdown induces fork stalling events.

4) In Fig S3a-b the authors show important controls for PLA specificity. It would be helpful to also add a quantification of PLA changes upon treatments in non-S-phase cells, further providing evidence for effect specificity to actively replicating cells. Particularly, the data of Fig. 3b also indicates slightly increased PLA foci levels in some G1 or G2/M cells upon HU/H₂O₂.

Response: The plots in Fig. 3b,c show a significant increase in PLA foci in some G2 cells, and to a lesser extent in some G1 cells upon HU/H₂O₂ treatment. The PLA foci in G1 cells might reflect non-specific events. The PLA foci in G2 cells may correspond to TRC events in late S-phase, where the level EdU incorporation is very low and therefore such cells are indistinguishable from G2 cells. However, from the bottom scatter plot in Fig. 3b in the revised manuscript, it is clearly evident that the increase in number of PLA foci upon treatments is predominantly in early/mid S phase cells with a high EdU incorporation. This is also apparent in the scatter plot shown in Fig. R7., where we plot separately the numbers of PLA foci in G1-, S- and G2-phase cells. However, we are of the opinion that it is not necessary to show this plot in our manuscript as it is clearly evident from the plots in Fig. 3b that PLA changes in response to HU/H₂O₂ treatment are specific to actively-replicating cells.

Figure R7. Scatter plot of the number of PCNA/RNAPII PLA foci in individual G1, S, G2 cells for indicated conditions. Experimental condition are described in the legend of Fig. 3a,b in the revised manuscript. Horizontal lines indicate the median.

5) In Fig 3f it would help the readability to mark early vs late replicating genes in the figures or mention which ones belong to which category also in the figure description.

Response: We now show only DRIP-qPCR data for several genes known to form R-loops (please see Suppl. Fig. 3f,g and the text on page 6/7).

6) In the discussion the authors mention that in U2OS 20% of early S-phase origins reside within protein-coding genes and that the genome is pervasively transcribed (II 241-243). This alone is not sufficient to conclude a frequent occurrence of HO TRCs. The residence of an origin within a gene does not necessitate a HO conflict. CD orientation is possible and probably the more likely case based on observations for a co-directional bias of transcription and replication.

Response: We removed those statements from the Discussion section.

7) The authors discuss new results in the discussion section, which should be avoided.

Response: The data shown in Suppl. Fig. 6 are now described in the Results section (page 9) and presented as Fig. 5 and Suppl. Fig. 5. Original Suppl. Fig. 7 was removed from the manuscript.

REVIEWERS' COMMENTS

Reviewer #1 (Remarks to the Author):

The authors have adequately addressed my comments.

Reviewer #2 (Remarks to the Author):

The authors have satisfactorily addressed most of my comments. In light of the data presented in the revised manuscript, I believe this study provides a significant step forward in enhancing understanding of the role of endogenous oxidative stress in transcription-dependent replication stress.

Reviewer #3 (Remarks to the Author):

The authors have done a good job in addressing my remaining issues and this study should be published in a timely manner.